# When To Solve, When To Verify: Compute-Optimal Problem Solving and Generative Verification for LLM Reasoning

**Nishad Singhi**[*1]**, Hritik Bansal**[*2]**, Arian Hosseini**[*3,4]**,**
**Aditya Grover**[2]**, Kai-Wei Chang**[2]**, Marcus Rohrbach**[1]**, Anna Rohrbach**[1]
[1]TU Darmstadt, hessian.AI    [2]UCLA    [3]Google Deepmind    [4]Mila    [*]Equal Contribution

## Abstract

Scaling test-time compute has emerged as a key strategy for enhancing the reasoning capabilities of large language models (LLMs), particularly in tasks like mathematical problem-solving. A traditional approach, Self-Consistency (SC), generates multiple solutions to a problem and selects the most common answer via majority voting. Another common method involves scoring each solution with a reward model (verifier) and choosing the best one. Recent advancements in Generative Reward Models (GenRM) reframe verification as a next-token prediction task, enabling inference-time scaling along a new axis. Specifically, GenRM generates multiple verification chains-of-thought to score each solution. Under a limited inference budget, this introduces a fundamental trade-off: should you spend the budget on scaling solutions via SC or generate fewer solutions and allocate compute to verification via GenRM? To address this, we evaluate GenRM against SC under a *fixed inference budget*. Interestingly, we find that SC is more compute-efficient than GenRM for most practical inference budgets across diverse models and datasets. For instance, GenRM first matches SC after consuming up to $8\times$ the inference compute and requires significantly more compute to outperform it. Furthermore, we derive inference scaling laws for the GenRM paradigm, revealing that compute-optimal inference favors scaling solution generation more aggressively than scaling the number of verifications. Our work provides practical guidance on optimizing test-time scaling by balancing solution generation and verification. The code is available at https://github.com/nishadsinghi/sc-genrm-scaling.

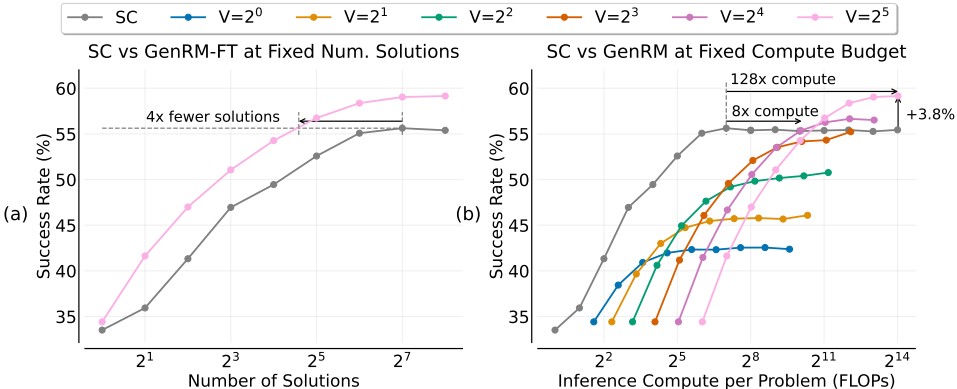

Figure 1: **Left:** The prominent approach is to compare GenRM and Self-Consistency (SC) at a fixed number of solutions, suggesting that GenRM is more efficient as it matches SC with fewer solutions. **Right:** When evaluated under a *fixed compute budget, including verification costs*, SC outperforms GenRM at lower budgets, using up to $8\times$ less compute, while GenRM excels at higher budgets. Each curve corresponds to a fixed number of verifications; the number of solutions is doubled at each point along the x-axis. The solutions are generated by Llama-3.1-8B-Instruct (Grattafiori et al., 2024), which also performs verifications after being fine-tuned as GenRM, on the MATH dataset (Hendrycks et al., 2021).

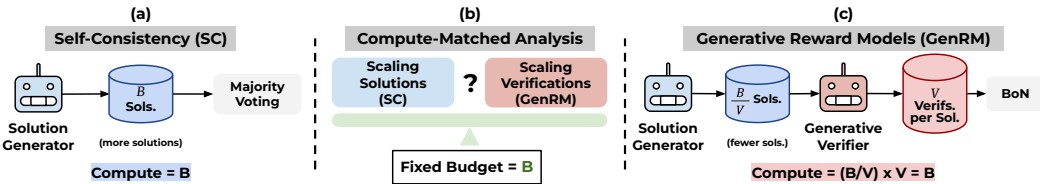

Figure 2: **Compute-Matched Analysis.** Given a fixed inference budget $B$, our analysis (b) compares the performance of (a) scaling the number of solutions ($S = B$) with *Self-Consistency* vs. (c) generating fewer solutions ($S = B/V$) while relying on verifications ($V$) using *Generative Reward Models*.

# 1   Introduction

Large Language Models (LLMs) have shown substantial improvements in their reasoning capabilities with increased test-time compute (Guo et al., 2025; Goyal et al., 2023; Snell et al., 2024) across diverse domains such as math and science. One of the most straightforward and effective ways to scale test-time compute is by generating multiple solution chains-of-thought (CoTs) for a given problem and performing majority voting over them, a technique known as *Self-Consistency (SC)* (Wang et al., 2022). Alternatively, a reward model (aka 'verifier') can be used to score the solutions and identify the best one. This strategy, commonly referred to as *Best-of-N (BoN)*, has been widely employed to enhance LLM reasoning capabilities (Cobbe et al., 2021). A robust reward model can detect errors and discard incorrect solutions, even when they are overrepresented among the generated solutions, making this approach particularly effective.

Recent studies (Zhang et al., 2024; Mahan et al., 2024; Ankner et al., 2024) have framed verification as a next-token prediction task, leading to a new class of reward models known as *Generative Reward Models (GenRM)*. These models enable test-time scaling along a new axis: generating multiple verification CoTs and aggregating their verdicts to score a given solution. Prominently, prior work compares the performance of GenRM (BoN) and SC at a fixed number of solutions. For instance, Llama GenRM (BoN) with 32 verification CoTs surpasses SC across different solution counts (Figure 1(a)). This comparison suggests that GenRM is more efficient, achieving the same performance as SC with $4\times$ fewer solutions. However, this conclusion is misleading in practical scenarios where inference compute budgets (FLOPs) are limited, as the current comparisons overlook the significant computational cost of generating many verifications for several candidate solutions of a given problem. This gives rise to the question: *At a fixed inference compute budget, is it more effective to perform SC over many solutions, or use GenRM to select the best solution from a smaller set of solutions by scaling the number of verifications?*

To address this, we present a framework to estimate the inference budget for Self-Consistency and GenRMs. Subsequently, we conduct a **compute-matched analysis**, comparing the effectiveness of these test-time scaling strategies under a fixed computational budget (§3.1). Specifically, we assume an LLM is used for both problem-solving (solution generator) and generative verification (reward model). The generative verification capability can be leveraged through either prompting or task-specific fine-tuning. Consequently, the inference compute comparison between SC and GenRMs is based on the total number of solutions and verifications generated by the LLM, as illustrated in Figure 2.

In our experiments, we compare the performance of scaling solutions and verifications with GenRM against simply scaling solutions via SC across various inference compute budgets. Our results indicate that **SC outperforms GenRM at lower compute budgets**, while **GenRM performs better at higher budgets** (Figure 1(b)). In particular, GenRM first surpasses SC after $8\times$ compute and requires an additional $128\times$ inference compute to achieve a 3.8% performance gain over SC. While prior work realized GenRM in limited settings, we demonstrate the **robustness of our findings** across various model families (e.g., Llama (Grattafiori et al., 2024) and Qwen (Yang et al., 2024)), model sizes (e.g., 7B and 70B), thinking models (e.g., QwQ-32B (Qwen, 2025)), and reasoning tasks (e.g., math) (§5.1).

As the compute budget is scaled and GenRM starts outperforming SC, a second challenge emerges: with GenRM, the available compute can be split between generating solutions and verifying them in different ways, leading to varying performance. For example, in Figure 1(b), using 8 verifications (red curve) performs better than 4 verifications (green curve) at a budget proportional to $2^8$ FLOPs. This shows a key tradeoff in GenRM: sampling too few solutions may lower the chances of generating a correct one (low coverage), while performing too few verifications can make it harder to identify the correct solution (low precision). Hence, this raises the question: *Under GenRM, how to allocate a given compute budget between generating solutions and verifying them to achieve optimal performance?*

To address this, we derive **inference scaling laws** for GenRM, which describe how the optimal number of solutions and verifications scales with the total compute budget (§3.2). We observe that while both need to be scaled in tandem, the **solutions should be scaled more rapidly than verifications**, by a factor of $1.5 - 2\times$, for optimal performance (§ 5.2). Overall, our work provides a solid foundation for understanding the trade-offs associated with test-time scaling of solutions and verifications, offering key insights for practitioners seeking to optimize their inference strategies and budgets.[1]

## 2 Background

**Repeated sampling.** A prominent approach for scaling test-time compute is repeated sampling, which involves generating multiple potential solutions from the LLM, $\mathcal{G}$, and selecting a final answer as the prediction. Specifically, we study two common methods for choosing the final answer: (a) **Self-Consistency (SC)** selects the most common answer—determined by majority voting—as the final answer (Wang et al., 2022). This strategy benefits from exploring diverse reasoning pathways and marginalizing over them, increasing the likelihood of correct answers. (b) **Best-of-N (BoN)** scores each candidate solution independently and selects the highest-scoring one (Cobbe et al., 2021). This method relies on a reward model capable of accurately assessing problem-solution pairs for correctness.

**Generative Verification.** Unlike traditional reward models that are discriminative, recent works have developed *Generative Reward Models (GenRM)* that pose verification as a next token prediction task (Zhang et al., 2024; Mahan et al., 2024). Concretely, the verifier takes as input the problem and the step-by-step solution, and provides a verification chain-of-thought (CoT) followed by its verdict (e.g., 'Is this answer correct? Yes/No'). This approach enables GenRM to inherit all the advantages of LLM reasoning, most notably test-time scaling, along a new axis: *verification*. In particular, we can sample multiple verification CoTs for each solution, and average over their verdicts to obtain a more accurate score.[2]

**GenRM-Base.** The simplest form of GenRM involves prompting an instruction-tuned LLM to verify a solution step-by-step and predict whether the candidate solution/answer is correct for the given problem. We refer to this as *GenRM-Base* since it uses an off-the-shelf LLM without specialized fine-tuning to act as GenRM.[3]

**GenRM-Finetuning.** Zhang et al. (2024) train an LLM to perform generative verification via supervised fine-tuning, which we denote as *GenRM-FT*. We start with a solution generator $\mathcal{G}$, a student verifier $r_{\text{student}}$ which will be fine-tuned to obtain GenRM-FT, and a teacher verifier $r_{\text{teach}}$ which is used to generate synthetic data to fine-tune $r_{\text{student}}$. The fine-tuning data is generated in the following way. We take a training dataset, $\mathcal{D}_{\text{train}} = \{(\mathbf{x}_i, \mathbf{y}_i, \mathbf{a}_i)\}$, consisting of problems $\mathbf{x}_i$, ground-truth solutions $\mathbf{y}_i$, and ground-truth answers $\mathbf{a}_i$. We use $\mathcal{G}$ to generate $N_s$ solutions for each problem in the dataset, where $\hat{\mathbf{y}}_{i,j}$ and $\hat{\mathbf{a}}_{i,j}$ are the $j^{\text{th}}$ solution and answer generated by $\mathcal{G}$ for problem $i$. Then, we use the teacher verifier $r_{\text{teach}}$ (usually a strong model like GPT-4o (Hurst et al., 2024)) to generate $N_v$ synthetic verification

---

[1]We will release the data, model, and code in the camera-ready version.

[2]While other sophisticated inference strategies exist (e.g. process-level scoring or tree search (Wu et al., 2024; Snell et al., 2024)), their practical use is constrained by the necessity of high-quality process-level training data and the inherent complexities associated with their training.

[3]This is akin to self-verification in Zhao et al. (2025) where an LLM verifies its own solutions.

rationales for the generated solutions $\hat{\mathbf{y}}_{i,j}$. The teacher verifier has access to the ground-truth solutions in its prompt, allowing it to generate accurate, high-quality verifications. Every verification generated by the teacher consists of chain-of-thought (CoT) reasoning about the solution's correctness, followed by a final verdict ('Yes' or 'No'). We filter these synthetic verifications, retaining only those whose final verdict matches the ground-truth correctness of the generated answer $\hat{\mathbf{a}}_{i,j}$. Further, we balance this data to have an equal number of 'Yes' and 'No' verifications, leading to the final fine-tuning dataset $\mathcal{D}_{\text{GenRM-FT}} = \{(\mathbf{v}_{\text{CoT}}, \mathbf{v} \mid \mathbf{x}, \hat{\mathbf{y}})\}$ consisting of verification rationales $\mathbf{v}_{\text{CoT}}$ and final verdicts $\mathbf{v}$. Finally, we fine-tune $r_{\text{student}}$ on this dataset to obtain GenRM-FT. During inference, we generate a verification rationale via the GenRM (-base or -FT) and use the probability of the 'Yes' token as the final score $r_{\text{GenRM}}(\mathbf{x}, \mathbf{y}) = p_\theta(\text{Yes}|\mathbf{x}, \mathbf{y}, \mathbf{v}_{\text{CoT}})$.[4]

**Test-time scaling with GenRM.** For every problem $\mathbf{x}_i$ in the test dataset, we generate $S$ samples $\{(\hat{\mathbf{y}}_{i,j}, \hat{\mathbf{a}}_{i,j})_{j=1}^{j=S}\}$ using the solution generator $\mathcal{G}$, where $\hat{\mathbf{y}}_{i,j}$ and $\hat{\mathbf{a}}_{i,j}$ refer to the $j^{\text{th}}$ solution and answer, respectively. For every problem-solution pair, we generate $V$ verifications $\{(\hat{\mathbf{v}}_{i,j,k}, \hat{r}_{i,j,k})_{k=1}^{k=V}\}$ using the generative verifier, where $\hat{r}_{i,j,k} \sim r_{\text{GenRM}}(\mathbf{x}_i, \hat{\mathbf{y}}_{i,j}) \in [0, 1]$ is the verification score. We obtain the final score for the solution $\hat{\mathbf{y}}_{i,j}$ by averaging the verification scores from all $V$ verifications as $\bar{r}_{i,j} = (1/V)\sum_{k=1}^{V} \hat{r}_{i,j,k}$. Finally, we pick the solution with the highest score $\bar{r}_{i,j}$ as the final solution. To sum up, GenRM allows time-test scaling along a new dimension by increasing the number of verifications $V$ in addition to increasing the number of solutions $S$.

**Thinking Models.** An emerging approach to scaling test-time computation is training models to generate longer chains-of-thought (CoTs) involving deep thinking before producing a solution. These models, e.g., DeepSeek-R1 (DeepSeek-AI, 2025), QwQ-32B (Qwen, 2025), are trained using reinforcement learning and can leverage additional test-time compute for self-verification, reflection, and backtracking to arrive at a final solution. However, the amount of compute they allocate to find and verify a solution in their thought process remains uncontrollable. Moreover, their reasoning mechanisms are not yet well understood in the current literature. Here, we focus on complementary inference strategies, which can also benefit thinking models, rather than sequentially refining a single solution. We present detailed related work in Appendix A.

## 3 Methodology

### 3.1 Compute-Matched Analysis of Test-time Scaling Strategies

Given the option to allocate a fixed inference budget toward either scaling solutions via Self-Consistency or verifying them using GenRMs, it remains unclear which approach is compute-optimal for LLM reasoning. To address this question, we conduct a compute-matched analysis of their respective scaling behaviors. We consider an autoregressive LLM with $P$ parameters that will perform $2P$ FLOPs per output token during inference (Kaplan et al., 2020). Hence, the number of inference FLOPs for generating $T$ tokens is $2PT$.

Let the number of tokens required for generating a solution and verification be $T_S$ and $T_V$, respectively. Following Zhang et al. (2024), we use the same model for problem-solving and generative verification. For instance, one might use Llama-8B to generate solutions and a fine-tuned version of the same model as GenRM-FT (or the same model without fine-tuning as GenRM-Base). Hence, the number of model parameters for the solution generator and verifier is identical, say $P$. Thus, the total inference compute (FLOPs) required to solve a reasoning problem with $S$ solutions and $V$ verifications is $\boxed{2P(T_S S + T_V S V)}$. Further, we consider $T_V = \lambda T_S$ where $\lambda$ is the ratio of the number of tokens per verification and solution. In our analysis, we use the formula $\boxed{C(S, V) = S(1 + \lambda V)}$ to measure inference

---

[4]We find that counting the number of 'Yes' across multiple verifications also works well.

compute for simplicity, as it is proportional to the total inference FLOPs for a given LLM. For Self-Consistency (SC), we set the number of verifications to $V = 0$.

We evaluate SC by sampling $S$ solutions and performing majority voting over them, for all $S \in \mathcal{S} = \{2^0, 2^1, ..., 2^N\}$, where $2^N$ is the maximum number of solutions. Similarly, we evaluate GenRM by varying the number of solutions $S \in \mathcal{S}$ and verifications $V \in \mathcal{V} = \{2^0, 2^1, ..., 2^M\}$, where $2^M$ is the maximum number of verifications per solution. For every combination $S, V \in \{\mathcal{S} \times \mathcal{V}\}$, we sample the corresponding number of solutions and verifications, and pick the final answer via Best-of-N. We compare the final answers against the ground-truth to compute success rates (SR), and plot them against the total compute, $C(S, V)$. Thus we compare the performance of GenRM and SC at the same compute budget.

## 3.2 Inference Scaling Laws for GenRM

As an emerging paradigm, GenRM lacks formalized inference scaling laws. Scaling test-time compute with GenRM involves scaling two independent axes: solutions and verifications. Further, the same amount of compute can be allocated between these two in different ways, leading to different performance outcomes. Hence, understanding the tradeoff between scaling solutions and verifications is crucial for compute-optimal inference. To address this, we extend the approach from Chinchilla (Hoffmann et al., 2022) to inference-time scaling. Specifically, we follow these steps:

1. Compute the success rate $\mathrm{SR}_{s,v}$ for an increasing number of verifications $v$, while keeping the number of solutions per problem $s$ constant.
2. Plot $[\mathrm{SR}_{s,2^0}, \ldots, \mathrm{SR}_{s,2^M}]$ against the inference compute budget $C = s(1 + \lambda v)$, and generate such plots for all values of $s$.
3. Smooth and interpolate curves to obtain a mapping from inference compute to SR.
4. For each budget, determine the optimal number of solutions ($S_{\mathrm{opt}}$) that maximizes SR (Appendix J Fig. 10). This results in a trend of $S_{\mathrm{opt}}$ as a function of the inference budget $C$.
5. Fit a power law, $S_{\mathrm{opt}} \propto C^a$, to establish a relationship between the optimal number of solutions and the inference budget.
6. Similarly, repeat steps 1–5 for verifications to compute $V_{\mathrm{opt}} \propto C^b$.

A higher value of $a$ ($b$) indicates that as compute increases, the number of solutions (verifications) must be scaled more rapidly compared to the number of verifications (solutions). If $a = b$, it implies that solutions and verifications should be scaled at the same rate.

## 4 Experimental Setup

**Tasks.** We use MATH (Hendrycks et al., 2021), a dataset of high-school competition problems to evaluate mathematical reasoning. Additionally, we utilize the MATH train split to train GenRM-FT models. To evaluate generalization of GenRM-FT to harder math tasks, we use the AIME24 dataset (AoPS, 2024), which consists of advanced high-school math problems. Finally, to evaluate reasoning beyond math, we use the GPQA-Diamond dataset (Rein et al., 2024), which consists of problems pertaining to physics, chemistry, and biology. Following Brown et al. (2024), we perform all experiments on a subset of 128 problems randomly sampled from the MATH test set. Similarly for GPQA, we perform experiments on a subset of 64 problems randomly sampled from the diamond split.

**Models.** Following prior works (Brown et al., 2024; Mahan et al., 2024; Ankner et al., 2024), we perform our experiments with Llama-3.1-8B-Instruct (Grattafiori et al., 2024). To ensure coverage across model families and sizes, we also experiment with Qwen-2.5-7B-Instruct (Yang et al., 2024) and Llama-3.3-70B-Instruct on the MATH dataset. We sample solutions with a temperature of 0.7 and with a maximum of 1024 tokens.

**GenRM-FT.** We fine-tune Llama-3.1-8B-Instruct and Qwen-2.5-7B-Instruct models to serve as GenRM-FT. To create the GenRM fine-tuning data, we use the corresponding models to generate solutions to problems from the MATH training split. Then, we use a stronger model, GPT-4o (Hurst et al., 2024), to generate verification rationales for these solutions.

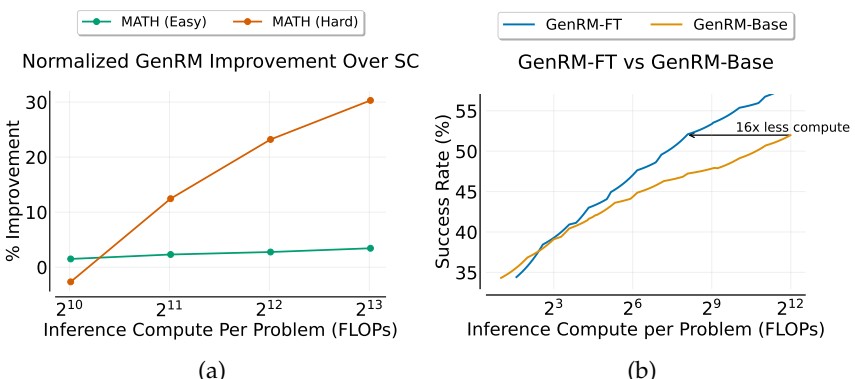

Figure 3: **Left:** Relative improvement achieved by Llama-3.1-8B-Instruct GenRM-FT (32 verifications) over SC for different difficulty levels in MATH. Hard problems benefit more from GenRM-FT, with up to 30% relative improvement over SC. **Right:** Comparing GenRM-FT against GenRM-Base, we find that GenRM-FT consistently performs better, requiring much less compute to match the performance of GenRM-Base. This highlights the importance of high-quality verifications.

Further details are available in Appendix C. We find that the verifications generated by GPT-4o tend to be lengthier than the solution itself, as it analyzes several steps in the solution before making its verdict (Appendix L). Hence, during inference, we sample up to 2048 tokens from our GenRM-FT models with a temperature of 0.7, i.e., $\lambda = 2$ for GenRM-FT. We sample up to 32 verifications for 256 solutions, i.e., $\mathcal{S} = \{2^0, ..., 2^8\}$ and $\mathcal{V} = \{2^0, ..., 2^5\}$.

**GenRM-Base.** We use Llama-3.3-70B-Instruct to generate solutions and verifications with no fine-tuning, i.e., GenRM-Base, due to its strong instruction-following and reasoning capabilities. We sample solutions and verifications using a temperature of 0.7 and a maximum length of 1024 tokens, i.e., $\lambda = 1$. We experiment with MATH and GPQA for this model.

**Evaluation.** We measure performance in terms of success rate (SR), i.e., the average percentage of problems solved on a test set. Following (Zhang et al., 2024; Hosseini et al., 2024), we use Best-of-N to select the final answer with a verifier. Unlike SC, Best-of-N can detect rare but correct solutions with an effective verifier (Zhao et al., 2025). We provide the verifier success rate in Appendix Eq. 1.

## 5 Experiments

First, we address the question whether at a given budget one should scale both solutions and verifications via GenRM or only scale solutions via Self-Consistency (SC). Hence, in §5.1, we compare the performance of these two approaches across a range of computational budgets. Further, using GenRM poses another question, as the same budget can be distributed between solutions and verifications in different ways, leading to different performance outcomes. Hence, we develop inference-time scaling laws for GenRM in §5.2.

### 5.1 Fixed Budget Comparison between Self-Consistency and GenRM

Following prior work (Zhang et al., 2024; Mahan et al., 2024), we compare the success rates of Self-Consistency (SC) and GenRM-FT (w/ 32 verifications) across different number of solutions using Llama-3.1-8B-Instruct on the MATH dataset in Figure 1(a). We see that GenRM-FT outperforms SC when both have the same number of solutions. Further, GenRM-FT matches the performance of SC with $4\times$ fewer solutions. However, this comparison does not account for the cost of generating verifications, and may give the misleading impression that GenRM is generally more efficient than SC.

For a fair comparison, we plot the success rates against the total compute used to generate solutions and verifications, and compare SC and GenRM at the same budget in Figure 1(b).

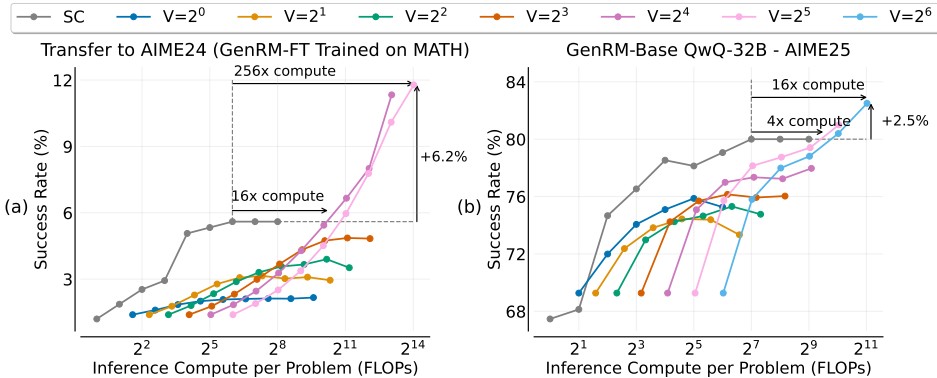

Figure 4: **(Left)** Evaluation of GenRM-FT (Llama-3.1-8B trained on MATH) generalizing to AIME24. GenRM-FT provides significant improvements over Self-Consistency (SC) on these harder problems, demonstrating its generalization ability, though it requires substantially more compute to outperform SC. **(Right)** Comparison of GenRM-Base versus SC for an RL-tuned QwQ-32B model. This confirms previous observations: SC performs better at lower budgets, while GenRM shines at higher budgets. We extrapolate the SC curves (dashed lines) because their performance saturates beyond a certain point.

Interestingly, we find that SC outperforms GenRM-FT at lower compute budgets. Notably, GenRM-FT requires $8\times$ more compute to match the performance of SC.

Similar to Brown et al. (2024), we find that the performance of SC plateaus at around 128 solutions, and sampling solutions further does not provide additional benefits. Consequently, allocating test-time compute to GenRM-FT only starts to yield benefits beyond a certain budget. Finally, we find that GenRM-FT achieves a 3.8% improvement over the best performance of SC, but requires $128\times$ more compute to achieve this performance.

> **Takeaway**
>
> At lower inference-compute budgets, scaling solutions using Self-Consistency leads to better performance than scaling both solutions and verifications with GenRM. However, at higher budgets, GenRM surpasses Self-Consistency in performance.

**Impact of Problem Difficulty.** When GenRM outperforms SC at higher inference budgets, we investigate which types of problems benefit the most. In Figure 3a, we analyze the relative improvement achieved by applying Best-of-N with GenRM-FT over Self-Consistency, computed as Improvement $= (\text{SR}_{\text{GenRM}} - \text{SR}_{\text{SC}})/\text{SR}_{\text{SC}}$. Specifically, we evaluate Llama-3.1-8B-Instruct on two difficulty levels from the MATH dataset: level 1 and level 5, which we denote as MATH (Easy) and MATH (Hard), respectively. Our results indicate that GenRM is particularly advantageous for more challenging problems, yielding up to a 30% relative improvement in performance. These findings can inform the choice between GenRM and SC, with GenRM offering greater benefits for harder problems.

**Impact of Verifier Quality.** We also compare the performance of GenRM-FT against GenRM-Base at a fixed compute budget in Figure 3b. We find that GenRM-FT consistently outperforms GenRM-Base, requiring up to $16\times$ less compute to reach the same performance. This highlights the benefit of fine-tuning LLMs for verification, especially for smaller models with weaker instruction-following and reasoning capabilities. Additionally, this suggests that as the verification capabilities of LLMs improve, GenRM based methods might become more compute-efficient. More details are available in Appendix E. We explore alternative approaches to improve GenRM efficiency in Appendix F.

**Easy-To-Hard Generalization.** In practice, GenRM-FT may encounter unseen problems with higher difficulty than the ones seen during training (Sun et al., 2024). Hence, we extend our analysis to a harder dataset (AIME-2024) for a GenRM-FT trained on an easier dataset (MATH). In particular, we compare SC and GenRM-FT for Llama-3.1-8B-Instruct in Figure 4(a). Interestingly, we find that our previous observations still hold: SC outperforms

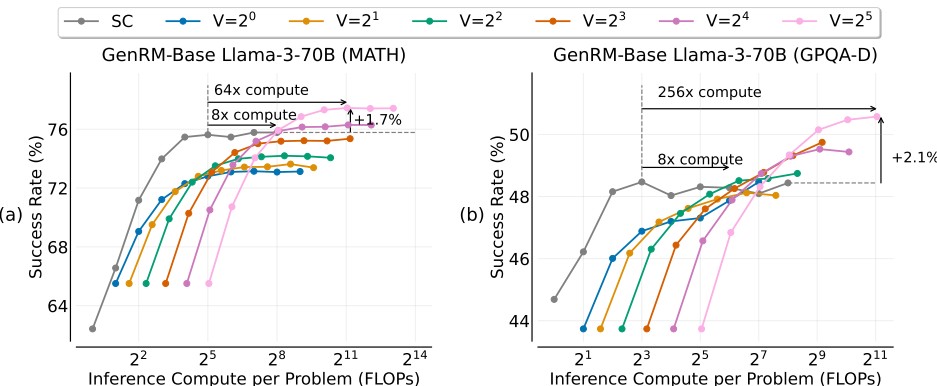

Figure 5: Comparing GenRM-Base with Llama-3.3-70B-Instruct on (a) MATH, and (b) GPQA-Diamond, respectively. These results highlight that across model sizes and reasoning domains, GenRM outperforms Self-consistency (SC) only at high compute budgets. We extrapolate the SC curve (horizontal dashed line) as its performance saturates after a point.

GenRM-FT at lower compute budgets, whereas GenRM-FT outperforms SC at higher budgets. For instance, SC achieves its peak performance using $16\times$ less inference compute than GenRM-FT needs to reach the same level. However, GenRM almost doubles the performance of SC, but requires $256\times$ more compute to do so. This highlights that generative verifiers can effectively generalize to much harder reasoning tasks.

**Trends Across Model Families.** To study whether our findings generalize to other model families, we compare GenRM-FT and SC with Qwen-2.5-7B-Instruct on the MATH dataset. Our results in Appendix Figure 9 are consistent with our previous findings: SC outperforms GenRM-FT for most of the lower end of the budget spectrum, achieving its peak performance with $64\times$ less compute than GenRM-FT. However, GenRM-FT shines at higher budgets, achieving an improvement of 5.4%, but by utilizing $512\times$ more compute.

**Trends for Thinking Models.** We extend our analysis to the emerging class of RL-tuned reasoning models that are capable of deep thinking involving self-reflection (e.g., 'aha moments') before generating the final solution (DeepSeek-AI, 2025). We evaluate QwQ-32B (Qwen, 2025) on the challenging AIME 2025 (AoPS, 2025) dataset as a solution generator and verifier (GenRM-Base). We provide more inference details in Appendix I. We present the results in Figure 4(b). Interestingly, we observe very similar trends for these models as well. In particular, GenRM requires $4\times$ more compute to match the performance of SC, and achieves a 2.5% improvement with $16\times$ more compute.

**Trends Across Model Sizes.** Here, we study whether our findings apply to larger LLMs. To this end, we experiment with Llama-3.3-70B-Instruct as GenRM-Base, which has $9\times$ more parameters than Llama-3-8B. The results in Figure 5(a) are consistent with our previous findings: SC performs better at lower budgets and achieves its peak performance with $4\times$ less compute as compared to GenRM-Base at the same performance. Further, GenRM-Base performs better at higher budgets, using $64\times$ more compute to improve performance by 1.7%. This also highlights that using strong models like Llama-3.3-70B as Generative Reward Models can improve reasoning performance even without fine-tuning them for verification.

**Trends Across Reasoning Domains.** Test-time scaling can benefit reasoning in domains beyond math, such as physics, chemistry, and biology. Hence, we compare SC against GenRM-Base at a fixed compute budget on GPQA-Diamond using Llama-3.3-70B-Instruct. Our results in Figure 5(b) show that GenRM-Base can provide a boost in reasoning abilities across domains, but only at a large budget. For instance, it uses $256\times$ more compute than SC to yield a 2.5% improvement on SC. At lower budgets, however, SC performs better. Additionally, we observe similar trends in the domain of code generation. Specifically, in Figure 8, we find that Llama 3 8B as GenRM-Base requires $16\times$ more compute to match SC and $64\times$ more compute to improve its performance by 3.5% on the LCB Benchmark (Jain et al., 2024).

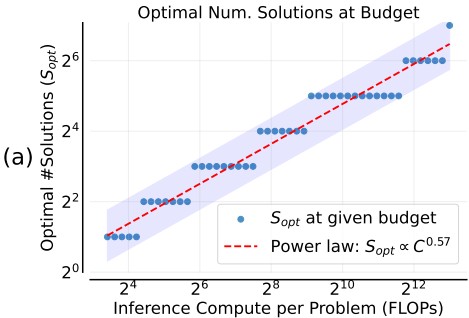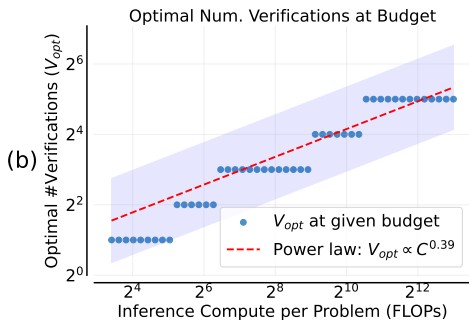

Figure 6: The optimal number of (left) solutions and (right) verifications for a given compute budget, using Llama-3-8B and GenRM-FT. Every point corresponds to a compute budget. The plots show that as the budget scales, the optimal number of solutions and verifications follows a power law, with the number of solutions increasing more rapidly. See Appendix J Fig. 10 for $S_{\mathrm{opt}}$ and $V_{\mathrm{opt}}$ computation.

> **Takeaway**
>
> Our findings about compute-optimality of SC and GenRM hold across model families, sizes, and tasks. Improving verification quality (GenRM-FT) benefits performance at fixed compute.

## 5.2 Inference Scaling Laws for Generative Reward Models

Our experiments so far have shown that GenRM becomes a favourable choice as the compute budget increases. However, it is important to strike a careful balance between the number of solutions and verifications to achieve optimal performance at a given budget. This raises the question: *What is the optimal way to allocate a given compute budget between generating solutions and verifying them?* To address this, we derive inference scaling laws, which describe how the optimal number of solutions and verifications scales with compute budget.

We sample up to 128 solutions per problem and up to 128 verifications per solution for the MATH test split using Llama-3.1-8B-Instruct and GenRM-FT. Then, we compute the performance at various values of $S$ and $V$ (Figure Appendix 11a) and identify the optimal number of solutions and verifications for a given budget. Subsequently, we study how the optimal number of solutions and verifications must be scaled as the budget is increased by fitting power law curves. Our findings in Figure 6 show that the optimal number of solutions scales as $S_{\mathrm{opt}} \propto C^{0.57}$ while the optimal number of verifications scales as $V_{\mathrm{opt}} \propto C^{0.39}$. The larger exponent associated with $S_{\mathrm{opt}}$ indicates that while both solutions and verifications should be scaled in tandem, solutions should be scaled at a faster rate for compute-optimal performance. Further experiments in Appendix J show that this finding holds across models.

> **Takeaway**
>
> We derive inference scaling laws for generative reward models along two axes: the number of solutions to verify and the number of verifications per solution. For compute-optimal inference, the number of solutions should scale 1.5–2× faster than the number of verifications.

## 6 Discussion

In this work, we conduct a compute-matched comparison between self-consistency and generative reward models. Specifically, we analyze performance under a fixed inference FLOP budget. A relevant direction for future work is to extend this analysis to a fixed latency (tokens/sec) setting across diverse hardware (e.g., GPUs/TPUs, different VRAM configurations) and algorithmic approaches (e.g., batching, efficient attention). Additionally, our

compute budget calculations focus on the widely used decoder-only autoregressive large language models (LLMs). However, emerging LLM architectures, such as sub-quadratic models (Gu & Dao, 2023) and language diffusion models (Nie et al., 2025), present promising directions for future exploration. A key bottleneck is the current lack of competitive reasoning-capable LLMs in these architectures. Finally, we evaluate the Best-of-N strategy with outcome-based verification using a generative verifier. Future work could extend compute-matched analyses to alternative inference strategies, such as beam search or lookaround search (Snell et al., 2024). It is also important to note that self-consistency can only be applied to problems with a definitive final answer. Therefore, domains such as theorem proving may not be amenable to self-consistency, necessitating the use of alternative approaches like Best-of-N.

## 7   Conclusion

Generative Reward Models (GenRMs) introduce a novel approach to scaling test-time compute through verifications. While prior work demonstrates that scaling both solutions and verifications can surpass Self-Consistency (SC), it often overlooks verification costs. In this study, we investigate whether scaling verifications improves performance under a fixed budget. We find that SC outperforms GenRMs at lower budgets, whereas GenRMs excel at higher ones. Our conclusions regarding the compute-optimality of SC and GenRMs across different budgets remain robust across various model families (including thinking models), sizes, and reasoning tasks. Furthermore, we derive inference scaling laws to optimize budget allocation between solutions and verifications in GenRM. Overall, our findings provide practical guidance for compute-efficient scaling to achieve optimal performance.

## Acknowledgements

Hritik Bansal is supported in part by AFOSR MURI grant FA9550-22-1-0380. Nishad Singhi is supported by a LOEWEStart-Professur (LOEWE/4b//519/05.01.002-(0006)/94). Marcus Rohrbach is supported in part by an Alexander von Humboldt Professorship in Multimodal Reliable AI, sponsored by Germany's Federal Ministry for Education and Research. We thank Ashima Suvarna, Xueqing Wu, Hector Garcia Rodriguez, Jonas Grebe, Tobias Wieczorek, and the anonymous reviewers for their helpful comments. For compute, we gratefully acknowledge support from the hessian.AI Service Center (funded by the Federal Ministry of Education and Research, BMBF, grant no. 01IS22091) and the hessian.AI Innovation Lab (funded by the Hessian Ministry for Digital Strategy and Innovation, grant no. S-DIW04/0013/003).

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

## A  Related Work

**Test-Time Compute Scaling.**  Leveraging more test-time compute to improve the performance of LLMs has gained a lot of popularity. Recent studies have explored various methods to scale test-time compute. A widely recognized baseline technique is repeatedly sampling candidate solutions from a model to choose the most frequent answer (aka self-consistency or majority-voting) (Wang et al., 2022). However, recent studies are pushing beyond this, investigating methods that leverage LLMs to iteratively refine their generated outputs (Gou et al., 2024; Cook et al., 2024; Lee et al., 2025). Reasoning models, such as OpenAI o3 series (OpenAI, 2024) and DeepSeek R1 (DeepSeek-AI, 2025) have enabled sequential scaling of test-time compute by scaling the length of the generated CoT (rather than parallel scaling by generating multiple shorter candidate solutions). While these long CoTs may implicitly incorporate forms of reflection, verification, or refinement within their extended reasoning sequence, such models and previous studies do not primarily address the compute optimality of their proposed methods, the main focus of our investigation.

**Verification.** Another common method to scale inference-time compute is to score candidate solutions through verification, which can be achieved via several techniques. Traditionally, discriminative models are employed, trained via binary classification (Cobbe et al., 2021; Luo et al., 2024; Yu et al., 2024) or preferences (Hosseini et al., 2024; Yuan et al., 2024). Generative verifiers frame verification as a next-token-prediction task, enabling the use of CoT reasoning and another axis to increase inference-time compute, either with trained verifiers (Zhang et al., 2024; Mahan et al., 2024; Ankner et al., 2024) or simply via prompting an off-the-shelf LLM (aka LLM-as-a-judge) (Bai et al., 2022; Zheng et al., 2023a; Chen et al., 2025; Kim et al., 2024; Zheng et al., 2023b) or self-verification based sampling (Zhao et al., 2025). These studies evaluate candidate solutions at outcome level, rather than process level verification (Lightman et al., 2023; Wang et al., 2024). Discriminative verifiers have become less favored due to the inherent challenges in their training and their tendency to exhibit lower performance compared to generative approaches (Zhang et al., 2024). Existing verification-based scaling studies focus on improving accuracy but ignore the overall cost of adding more verifications. They are not concerned with the compute optimal setting about spending one's budget on generating more candidate solutions or more verifications for existing solutions.

**Inference Scaling Laws.** We can better allocate resources by understanding the characteristics of different inference scaling strategies. Snell et al. (2024) study how scaling test-time compute, through search against dense rewards and adaptive response updates impacts reasoning performance, revealing prompt difficultly as a key factor. In contrast, our work takes into account the verification budget and studies the trade-off between allocating compute to generate new candidate solutions versus verifying existing ones. Wu et al. (2024) investigate compute-optimal scaling, specifically examining the trade-off between model size and generating multiple samples. Consistent with findings in Wu et al. (2024); Brown et al. (2024); Bansal et al. (2024), they demonstrate that sampling multiple times from a smaller model can outperform a larger and stronger model within a fixed budget. However, we focus our scaling analysis on the trade-off between generating additional solution candidates and generating verifications for existing solutions. Setlur et al. (2025) argue that test-time compute scaling without verification is suboptimal which is consistent with our overall findings.

## B  Inference Details

Following Brown et al. (2024), we use vLLM (Kwon et al., 2023) with up to 16 A100 GPUs to generate solutions and verifications from LLMs.

**Math Datasets**   Following Brown et al. (2024), we use a 4-shot prompt to generate solutions (Prompt K.1). We use the `minerva_math` function from LMEval (Gao et al., 2023) to extract the final answer from the solution, and the `is_equiv` function to check if the answer matches the ground-truth answer. For GenRM-base, we use a 2-shot prompt to generate verifications (Prompt K.2). For GenRM-FT, we use Prompt K.3 for both fine-tuning and inference.

**GPQA** We perform our experiments on a subset of 64 randomly sampled problems from the diamond split of GPQA. We use a zero-shot prompt to generate solutions (Prompt K.5). We sample with a temperature of 0.7 and with the maximum number of tokens set to 1024. For GenRM-Base, we use a zero-shot prompt to generate verifications (Prompt K.6) with a temperature of 0.7 and a maximum of 1024 tokens.

## C GenRM Training Details

We use the solution generator, e.g., Llama-3.1-8B-Instruct, to generate 4 solutions for problems in the MATH training split (7500 problems), using the prompt described in Appendix B. Then, we provide these solutions to GPT-4o along with the ground-truth solutions, and generate 4 verifications for every solution using Prompt K.4. We filter out the verifications whose verdict doesn't match the ground-truth correctness of the solution. Further, we sample verifications such that the number of correct and incorrect samples is balanced. We use LoRA (Hu et al., 2022) to fine-tune the same model on this dataset of synthetic verification rationales for 3 epochs. We pick the learning rate from $\{5e-7, 1e-6, 1e-5\}$ based on accuracy on a validation split (10% of training split).

## D Evaluation Details

**Reliable Estimation of Best-of-N** Following Hosseini et al. (2024), we estimate the average success rate by estimating the probability that when sampling $k$ out of $N(>k)$ solutions, the one with the highest score is correct, and averaging it over $K$ repetitions:

$$\text{Best-of-}k := \frac{1}{\binom{N}{k}} \sum_{i=0}^{N-k} \binom{N-i-1}{k-1} \alpha_i \tag{1}$$

where $[\alpha_0, \alpha_1, ..., \alpha_{N-1}]$ are the binary correctness scores (0 or 1) for the candidate solutions sorted in decreasing order of their verifier scores.

## E Impact of Verifier Quality

Our results indicate that using a small number of verifications does not lead to competitive performance. In most cases, at least 32 verifications are needed to achieve good results, which significantly increases computational cost. This suggests that individual verifications may be noisy. If verification quality improves, however, then fewer verifications might be required to achieve the same performance.

To investigate this, we compare two generative verifiers: (1) a "strong" verifier: Llama-3.1-8B-Instruct fine-tuned on GPT-4o verification rationales (GenRM-FT), and (2) a "weak" verifier: Llama-3.1-8B-Instruct using a two-shot verification prompt (GenRM-base) on the MATH dataset. We use $\lambda = 1$ for GenRM-Base and $\lambda = 2$ for GenRM-FT.

Figure 3b shows how these verifiers perform as the number of solutions and verifications increases. For clarity, we only plot the best-performing configurations (number of solutions and verifications) for a given compute budget. Our findings reveal that across the board, the strong verifier achieves similar performance with significantly less (up to 16×) compute than the weak verifier. Moreover, the strong verifier outperforms the weak verifier overall.

These results underscore the importance of high-quality verification rationales. As the verification capabilities of LLMs improve in the future, the compute required for GenRM to achieve strong performance may decrease.

## F Improving Verifier Efficiency

In Section 5.1, we observed that improving the verification capabilities of GenRM via fine-tuning reduces its inference compute requirements, thereby enhancing efficiency. In this

section, we conduct preliminary explorations of alternative ways to improve GenRM's efficiency. Specifically, we focus on two approaches to reduce inference compute: using smaller verifiers and training verifiers to generate fewer tokens.

First, we use LLaMA 3 70B to generate solutions on the MATH benchmark. Keeping these solutions fixed, we consider two verifiers: LLaMA 3 70B as GenRM-Base and LLaMA 3 8B as GenRM-FT. As discussed in Section 4, GenRM-Base has $\lambda = 1$, so the compute required to generate a verification using GenRM-Base is equal to that required to generate a solution. In contrast, the smaller GenRM-FT has $\lambda = 2$, meaning the compute required to generate a verification is $2 \times 8/70 = 0.228$ times that required to generate a solution. Thus, generating verifications using GenRM-FT is significantly cheaper than generating solutions. We compare these two verifiers against SC in Figure 7. We find that while the smaller verifier reduces the compute needed to match SC from $8\times$ to $4\times$, it still does not outperform SC at low compute budgets.

As discussed in Section 3, the inference compute for verification is proportional to the number of tokens generated. Hence, we can further reduce the compute required for verification—and consequently for GenRM—by generating fewer verification tokens. Next, we analyze the impact of improving verifier efficiency through shorter verification outputs.

To estimate the upper bound on potential efficiency gains from generating fewer tokens, we vary the value of $\lambda$ while assuming constant verification quality. Note that this assumption is made solely for analytical purposes; in practice, training verifiers that maintain the same accuracy while generating fewer tokens is non-trivial. In Figure 1, we used LLaMA 3 8B as GenRM-FT on the MATH benchmark, where each verification uses 2048 tokens on average. In Table 1, we compare GenRM against SC while varying the number of verification tokens, assuming constant verification performance. We find that GenRM would need to reduce the number of verification tokens from 2048 to 256 (i.e., an $8\times$ reduction) to match the compute requirements of SC. Note that here we use the same model sizes (8B parameters) for generating solutions and verifications and only change the number of verification tokens.

The results in these sections show that training verifiers that are smaller and generate fewer verification tokens is a promising direction to improve the efficiency of GenRM. However, further research is required to bridge the gap between GenRM and SC.

Table 1: Comparison of GenRM performance against SC with varying values of verification tokens. We find that current GenRMs would have to generate $8\times$ fewer tokens, while maintaining their performance, to match the performance of SC.

| Tokens per verification | Compute to match SC | Compute to outperform SC |
|---|---|---|
| 2048 | $8\times$ | $128\times$ |
| 1024 | $4\times$ | $64\times$ |
| 512 | $2\times$ | $32\times$ |
| 256 | $1\times$ | $16\times$ |

## G Results on Code Generation

In this section, we extend our comparison between Self-consistency and GenRM to code generation tasks. We note that it is not straightforward to apply Self-consistency to code generation because the task is to generate an entire piece of code, not just a final answer. To apply Self-consistency to this task, we follow the methodology of Chen et al. (2023) and perform majority voting over the outputs of the generated code for the given test inputs. Specifically, we use Llama 3.1 8B Instruct to generate solutions and verifications as GenRM-Base on 128 problems from the LiveCodeBench (LCB) benchmark (Jain et al., 2024). Our results in Figure 8 show that GenRM requires $16\times$ more compute to match the performance of SC, and $64\times$ more compute to achieve a 3.5% improvement.

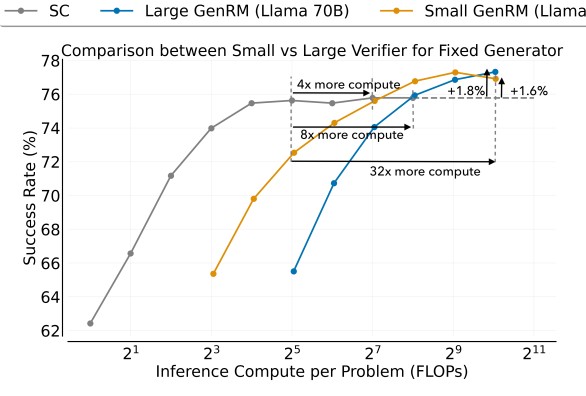

Figure 7: Comparing small (Llama 3 8B as GenRM-FT) and large (Llama 3 70B as GenRM-Base) verifiers against SC on MATH. While the smaller verifier is more efficient than the larger verifier, it still does not outperform SC at lower budgets.

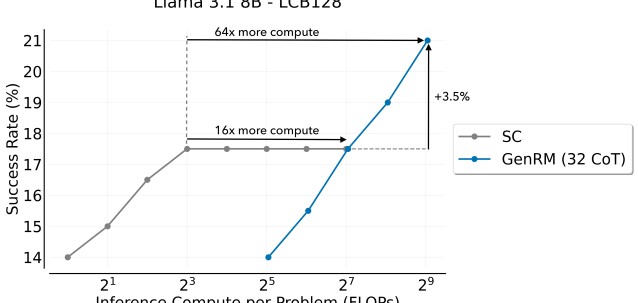

Figure 8: Comparing GenRM-Base against Self-consistency (SC) with Llama-3.1-8B-Instruct. SC is better at lower budgets while GenRM outperforms at higher budgets, suggesting that our findings generalize to code generation.

## H    Trends Across Model Families

In Figure 9, we present a compute-matched comparison between SC and GenRM-FT with Qwen-2.5-7B-Instruct on the MATH dataset. Similar to previous observations, we find that SC outperforms GenRM-FT at lower budgets, whereas GenRM shines at higher budgets.

## I    Inference Details For Thinking Model

In this work, we evaluate QwQ-32B on the challenging AIME2025 benchmark.[5] We use Prompt K.7 (Guha et al., 2024) and Prompt K.8 to generate solutions and verifications, respectively. Since this model generates much longer reasoning traces than traditional instruction-tuned models, we generate a maximum of 32768 tokens ($\lambda = 1$) with a temperature of 0.7 for both solution and verification tasks. We note that the compute required to generate the thought process within the <think> tokens is included in the solution and verification token budget.

## J    Additional Details and Results on Compute-Optimal Scaling Analysis

In this section, we derive scaling laws for Qwen-2.5-7B-Instruct and Llama-3.3-70B-Instruct on the MATH dataset. For Qwen-2.5-7B-Instruct, we sample 128 solutions and 128 verifications using GenRM-FT. Our findings in Figure 12 show that as the budget is increased, the optimal number of solutions scales as $S_{\text{opt}} \propto C^{0.75}$ and $V_{\text{opt}} \propto C^{0.32}$. For Llama-3.3-70B-Instruct, we sample 64 solutions and 64 verifications using GenRM-Base. Figure 13 shows that $S_{\text{opt}} \propto C^{0.69}$ and $V_{\text{opt}} \propto C^{0.43}$. These results show that as test-time compute is scaled, the number of solutions should be scaled more rapidly as compared to the number of verifications.

---

[5]We observed significant performance drop on AIME2025 in comparison to MATH and AIME2024. This highlights that the scope of improvement is higher with verification in comparison to saturated datasets.

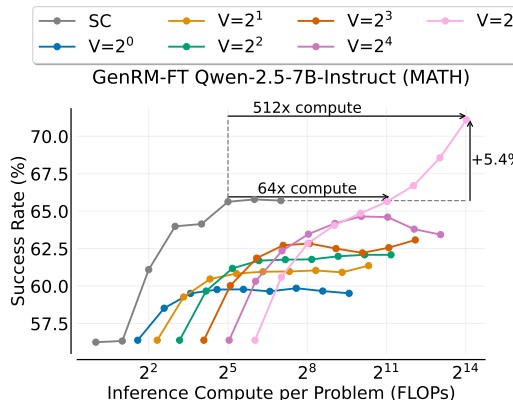

Figure 9: Comparing GenRM-FT against Self-consistency (SC) with Qwen-2.5-7B-Instruct. SC is better at lower budgets while GenRM outperforms at higher budgets, suggesting that our findings hold across model families.

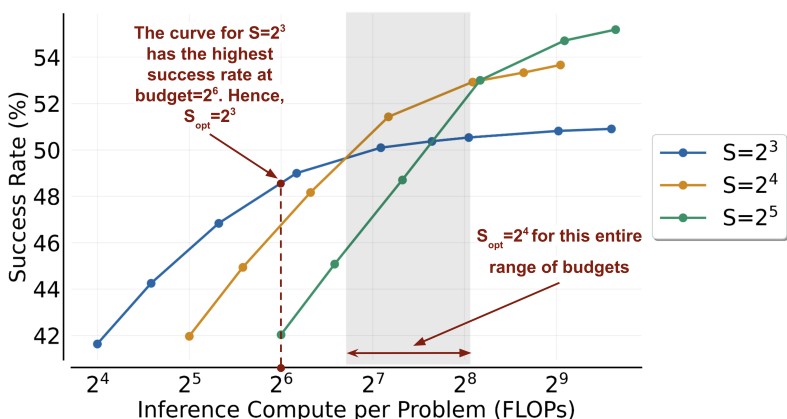

Figure 10: Toy illustration of how we compute $S_{\text{opt}}$ for a given budget. Every curve corresponds to a fixed number of solutions, with the number of verifications increasing along x-axis. For any budget on the x-axis, we find the curve that has the highest success rate at that budget. For instance, at a budget of $2^6$, the curve of $S = 2^3$ has the highest success rate, hence, $S_{\text{opt}} = 2^3$ at this budget. Further, the value of $S_{\text{opt}}$ increases in step-changes, as the number of optimal solutions must be an integer. The figure shows that $S_{\text{opt}} = 2^4$ for budgets between $\approx 2^7$ and $\approx 2^8$. $V_{\text{opt}}$ can be computed analogously from a plot of fixed verifications and increasing number of solutions.

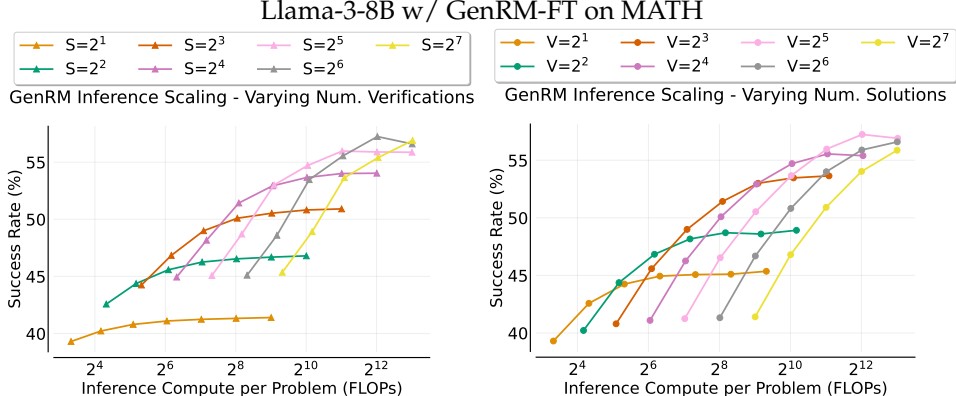

(a) Scaling trends of GenRM at (Left) a fixed number of solutions and increasing the number of verifications, and (Right) a fixed number of verifications and increasing the number of solutions.

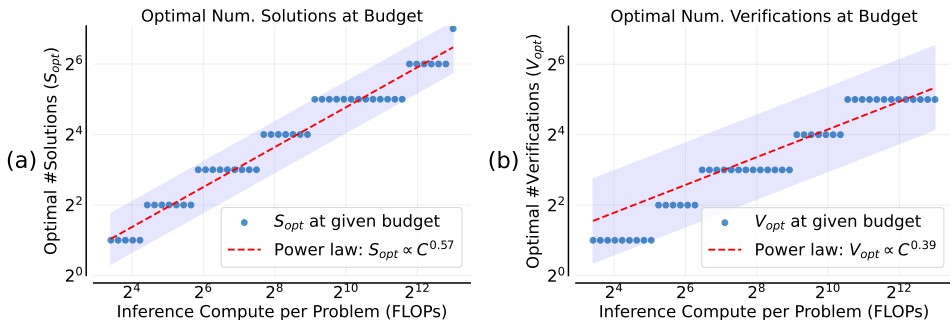

(b) The optimal number of (a) solutions and (b) verifications for a given compute budget. Every point corresponds to a compute budget. The plots show that as the budget scales, the optimal number of solutions and verifications follows a power law, with the number of solutions increasing more rapidly.

Figure 11: Compute-optimal scaling of solutions and verifications in GenRM-FT with Llama-3.1-8B-Instruct on MATH.

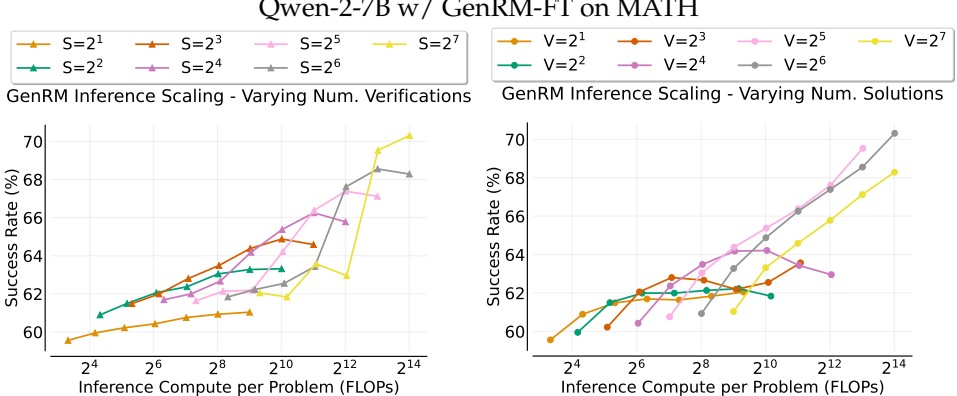

(a) Scaling trends of GenRM at (Left) a fixed number of solutions and increasing the number of verifications, and (Right) a fixed number of verifications and increasing the number of solutions.

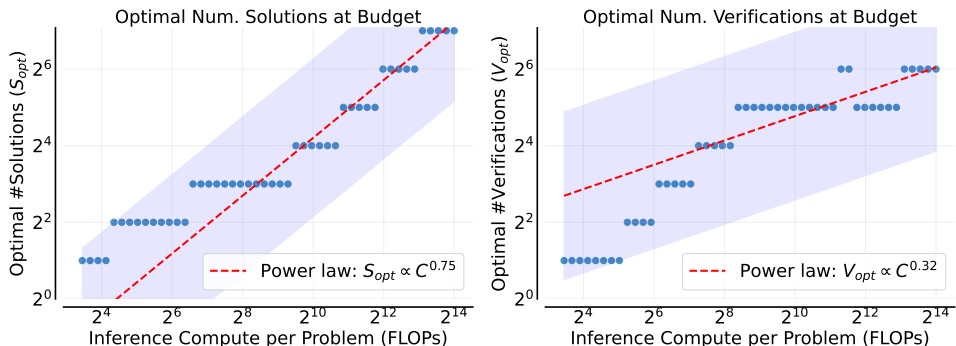

(b) The optimal number of (a) solutions and (b) verifications for a given compute budget. Every point corresponds to a compute budget. The plots show that as the budget scales, the optimal number of solutions and verifications follows a power law, with the number of solutions increasing more rapidly.

Figure 12: Compute-optimal scaling of solutions and verifications in GenRM-FT with Qwen-2.5-7B-Instruct on MATH.

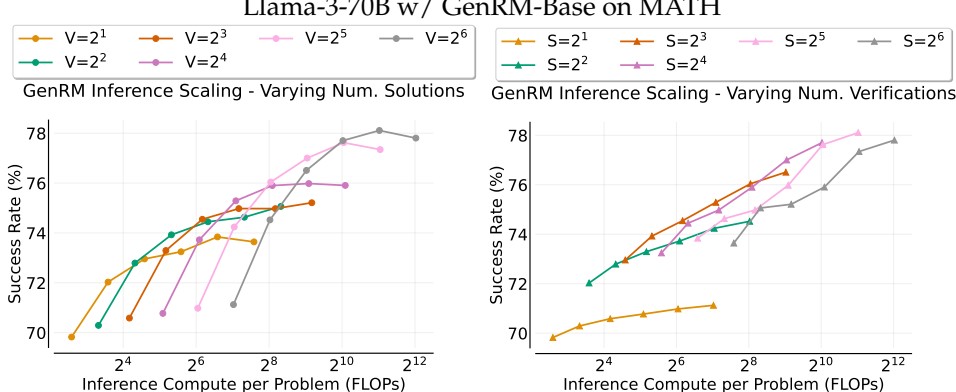

(a) Scaling trends of GenRM at (Left) a fixed number of solutions and increasing the number of verifications, and (Right) a fixed number of verifications and increasing the number of solutions.

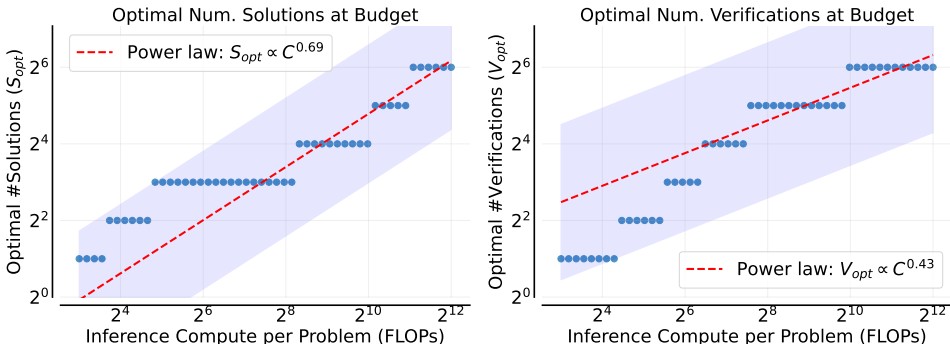

(b) The optimal number of (a) solutions and (b) verifications for a given compute budget. Every point corresponds to a compute budget. The plots show that as the budget scales, the optimal number of solutions and verifications follows a power law, with the number of solutions increasing more rapidly.

Figure 13: Compute-optimal scaling of solutions and verifications in GenRM-Base with Llama-3.3-70B-Instruct on MATH.

# K Prompts

## K.1 Generating Solutions to Math Problems

---

**4-shot Prompt for generating solutions to math problems**

```
1   Problem:
2   Find the domain of the expression  $\\frac{\\sqrt{x-2}}{\\sqrt
    {5-x}}$.}
3
4   Solution:
5   The expressions inside each square root must be non-negative.
    Therefore, $x-2 \\ge 0$, so $x\\ge2$, and $5 - x \\ge 0$, so $x
    \\le 5$. Also, the denominator cannot be equal to zero, so $5-x
    >0$, which gives $x<5$. Therefore, the domain of the expression
    is $\\boxed{[2,5)}$.\nFinal Answer: The final answer is $[2,5)$.
     I hope it is correct.
6
7   Problem:
8   If $\\det \\mathbf{A} = 2$ and $\\det \\mathbf{B} = 12,$ then
    find $\\det (\\mathbf{A} \\mathbf{B}).$
9
10  Solution:
11  We have that $\\det (\\mathbf{A} \\mathbf{B}) = (\\det \\mathbf{
    A})(\\det \\mathbf{B}) = (2)(12) = \\boxed{24}.$\nFinal Answer:
    The final answer is $24$. I hope it is correct.
12
13  Problem:
14  Terrell usually lifts two 20-pound weights 12 times. If he uses
    two 15-pound weights instead, how many times must Terrell lift
    them in order to lift the same total weight?
15
16  Solution:
17  If Terrell lifts two 20-pound weights 12 times, he lifts a total
     of $2\\cdot 12\\cdot20=480$ pounds of weight.  If he lifts two
    15-pound weights instead for $n$ times, he will lift a total of
    $2\\cdot15\\cdot n=30n$ pounds of weight.  Equating this to 480
    pounds, we can solve for $n$:\n\\begin{align*}\n30n&=480\\\n\\
    Rightarrow\\qquad n&=480/30=\\boxed{16}\n\\end{align*}\nFinal
    Answer: The final answer is $16$. I hope it is correct.
18
19  Problem:
20  If the system of equations\n\n\\begin{align*}\n6x-4y&=a,\\\n6y-9
    x &=b.\n\\end{align*}has a solution $(x, y)$ where $x$ and $y$
    are both nonzero,\nfind $\\frac{a}{b},$ assuming $b$ is nonzero.
21
22  Solution:
23  If we multiply the first equation by $-\\frac{3}{2}$, we obtain\
    n\n$$6y-9x=-\\frac{3}{2}a.$$Since we also know that $6y-9x=b$,
    we have\n\n$$-\\frac{3}{2}a=b\\Rightarrow\\frac{a}{b}=\\boxed
    {-\\frac{2}{3}}.$$\nFinal Answer: The final answer is $-\\frac
    {2}{3}$. I hope it is correct.
24
25  Problem:
26  <Problem>
27
28  Solution:
```

---

## K.2 GenRM-Base

---

**Few-shot GenRM-Base prompt for math problems**

```
1    Problem:
2    Find the domain of the expression  $\\frac{\\sqrt{x-2}}{\\sqrt
     {5-x}}$.}
3
4    Solution:
5    The expressions inside each square root must be non-negative.
     Therefore, $x-2 \\ge 0$, so $x\\ge2$, and $5 - x \\ge 0$, so $x
     \\le 5$. Also, the denominator cannot be equal to zero, so $5-x
     >0$, which gives $x<5$. Therefore, the domain of the expression
     is $\\boxed{[2,5)}$.\nFinal Answer: The final answer is $[2,5)$.
      I hope it is correct.
6
7    Problem:
8    If $\\det \\mathbf{A} = 2$ and $\\det \\mathbf{B} = 12,$ then
     find $\\det (\\mathbf{A} \\mathbf{B}).$
9
10   Solution:
11   We have that $\\det (\\mathbf{A} \\mathbf{B}) = (\\det \\mathbf{
     A})(\\det \\mathbf{B}) = (2)(12) = \\boxed{24}.$\nFinal Answer:
     The final answer is $24$. I hope it is correct.
12
13   Problem:
14   Terrell usually lifts two 20-pound weights 12 times. If he uses
     two 15-pound weights instead, how many times must Terrell lift
     them in order to lift the same total weight?
15
16   Solution:
17   If Terrell lifts two 20-pound weights 12 times, he lifts a total
      of $2\\cdot 12\\cdot20=480$ pounds of weight.  If he lifts two
     15-pound weights instead for $n$ times, he will lift a total of
     $2\\cdot15\\cdot n=30n$ pounds of weight.  Equating this to 480
     pounds, we can solve for $n$:\n\\begin{align*}\n30n&=480\\\n\\
     Rightarrow\\qquad n&=480/30=\\boxed{16}\n\\end{align*}\nFinal
     Answer: The final answer is $16$. I hope it is correct.
18
19   Problem:
20   If the system of equations\n\n\\begin{align*}\n6x-4y&=a,\\\n6y-9
     x &=b.\n\\end{align*}has a solution $(x, y)$ where $x$ and $y$
     are both nonzero,\nfind $\\frac{a}{b},$ assuming $b$ is nonzero.
21
22   Solution:
23   If we multiply the first equation by $-\\frac{3}{2}$, we obtain\
     n\n$$6y-9x=-\\frac{3}{2}a.$$Since we also know that $6y-9x=b$,
     we have\n\n$$-\\frac{3}{2}a=b\\Rightarrow\\frac{a}{b}=\\boxed
     {-\\frac{2}{3}}.$$\nFinal Answer: The final answer is $-\\frac
     {2}{3}$. I hope it is correct.
24
25   Problem:
26   <Problem>
27
28   Solution:
```

---

### K.3 GenRM-FT

---
**Prompt used for GenRM-FT**

```
1    Problem:
2    {}
3
4    Solution:
5    {}
6
7    Verification:
```
---

### K.4 Prompt to Generate Synthetic Training Data

---
**Prompt used to generate synthetic verification rationales for training GenRM**

```
1    You are a math teacher. Grade the Solution, verifying
     correctness step by step. At the end of the Solution
     verification, when you give your final grade, write it in the
     form 'Verification: Is the answer correct (Yes/No)? X', where X
     is either Yes or No.
2
3    Example 1:
4
5    Question:
6    <Question>
7
8    Solution:
9    <Solution>
10
11   Expected Solution:
12   <Ground-truth Solution>
13
14
15   Teacher Verification:
16   <Ground-truth verification rationale>
17
18   Verification: Is the answer correct (Yes/No)? No
19
20   ---
21
22   Example 2:
23
24   Question:
25   <Question>
26
27   Solution:
28   <Solution>
29
30   Expected Solution:
31   <Ground-truth Solution>
32
33
34   Teacher Verification:
35   <Ground-truth verification rationale>
36
37   Verification: Is the answer correct (Yes/No)? No
38
39   --
```
---

```
40
41    Now, continue grading the next solution step-by-step as follows.
42
43    Question: {}
44
45    Solution: {}
46
47    Expected Solution: {}
48
49    Teacher Verification:
```

## K.5   GPQA Solution

**Prompt used to generate solutions to GPQA problems**

```
1    Think step-by-step to solve the following problem. Only include
     the letter choice
2    (A, B, C, or D) as your final response. End your answer with ''
     Final Answer: The
3    final answer is X. I hope it is correct.'', where X is your
     final answer.
4
5    Problem: <Problem>
6
7    Options: <Options>
8
9    Answer:
```

## K.6   GPQA Verification

**Prompt used to generate verifications to GPQA solutions with GenRM-Base**

```
1    You are a teacher expert in physics, biology, and chemistry.
     Grade the Solution, verifying correctness step by step. At the
     end of the Solution verification, when you give your final grade
     , write it in the form 'Verification: Is the answer correct (Yes
     /No)? X', where X is either Yes or No.
2
3    Question: {}
4
5    Solution: {}
6
7    Teacher Verification:
```

## K.7   Thinking Models Solution Prompt

**Prompt used to generate solutions from reasoning models**

```
1    Problem: {}\n\
2    Mark your solution with \\boxed\nAnswer:
```

## K.8 Thinking Models Verification Prompt

> **Prompt used to generate verifications from reasoning models as GenRM-Base**
>
> ```
> 1   **Question:** {}
> 2
> 3   **Student Solution:** {}
> 4
> 5   # Math Solution Grading Instructions
> 6
> 7   As a math teacher, your role is to grade student solutions
>     following these standardized steps:
> 8
> 9   ## Verification Process
> 10  1. Perform a detailed verification by:
> 11     - Checking each step sequentially
> 12     - Verifying all calculations
> 13     - Validating mathematical properties and rules used
> 14     - Examining the logical flow between steps
> 15     - Confirming proper notation and mathematical writing
> 16
> 17  ## Error Documentation
> 18  2. For any errors found:
> 19     - Point out the specific location of the error
> 20     - Explain why it's incorrect
> 21     - Show the correct approach
> 22     - Note if the error affected subsequent steps
> 23
> 24  ## Assessment Format
> 25  3. End your answer in this format:
> 26  "Verification: Is the answer correct (Yes/No)? X". Where X is
>     Yes or No.
> 27
> 28  **Verification:**
> 29  Alright, let's verify the solution step by step.
> ```

# L   Verification Example from GPT-4o

> **Verification Example from GPT-4o**
>
> ```
> 1   Problem:
> 2   A street has 20 houses on each side, for a total of 40 houses.
>     The addresses on the south side of the street form an arithmetic
>      sequence, as do the addresses on the north side of the street.
>     On the south side, the addresses are 4, 10, 16, etc., and on the
>      north side they are 3, 9, 15, etc. A sign painter paints house
>     numbers on a house for $\$1$ per digit. If he paints the
>     appropriate house number once on each of these 40 houses, how
>     many dollars does he collect?
> 3
> 4   Solution:
> 5   There are $20$ houses on each side, so the painter will paint a
>     total of $40$ numbers. The house numbers on the south side are
>     $4, 10, 16, 22, 28, 34, 40, 46, 52, 58, 64, 70, 76, 82, 88, 94,
>     100, 106, 112, 118$ and the house numbers on the north side are
>     $3, 9, 15, 21, 27, 33, 39, 45, 51, 57, 63, 69, 75, 81, 87, 93,
>     99, 105, 111, 117$. Among the house numbers on the south side,
>     the number of digits in each number is as follows:
> ```

```
 6    - 4 and 40 have 1 digit.
 7    - 10, 22, 28, 34, 46, 52, 58, 64, 70, 76, 82, 88, 94, 100, 106,
      112, 118 have 2 digits.
 8    - 16, 26, 36, 46, 56, 66, 76, 86, 96 have 2 digits.
 9    - 58, 68, 78, 88, 98 have 2 digits.
10    - 64, 74, 84, 94 have 2 digits.
11    - 100, 110 have 3 digits.
12    Among the house numbers on the north side, the number of digits
      in each number is as follows:
13    - 3 and 33 and 63 and 93 and 123 have 1 digit.
14    - 9, 21, 39, 51, 69, 81, 99 have 1 digit.
15    - 15, 27, 45, 57, 75, 87, 105 have 2 digits.
16    - 23, 35, 47, 59, 71, 83, 95, 107 have 2 digits.
17    - 25, 37, 49, 61, 73, 85, 97 have 2 digits.
18    - 29, 41, 53, 65, 77, 89, 101, 113 have 2 digits.
19    - 43, 55, 67, 79, 91, 103, 115 have 2 digits.
20    - 55, 67, 79, 91, 103, 115 have 2 digits.
21    - 119 has 3 digits.
22    Since there are 1 one-digit number, 7 two-digit numbers, and 1
      three-digit number among the house numbers on the south side,
      the total amount of money the painter will collect is $(1\cdot
      1)+(2\cdot 17)+(3\cdot 1)=\boxed{39}$ dollars. Similarly, since
      there are 5 one-digit numbers, 7 two-digit numbers, 1 three-
      digit number among the house numbers on the north side, the
      total amount of money the painter will collect is $(1\cdot 5)
      +(2\cdot 7)+(3\cdot 1)=\boxed{22}$ dollars. Therefore, the total
       amount of money the painter will collect is 39+22=61 dollars.
23    Final Answer: The final answer is 61. I hope it is correct.
24
25
26
27    Verification: Let's verify the student's solution step by step
      against the expected solution:
28
29    ---
30
31    ### **Step 1: Problem Understanding**
32
33    The problem asks for the total amount of money collected by a
      sign painter to paint house numbers on 40 houses, divided into
      20 houses on the north and south sides of the street. Each house
       has a unique address, determined by the arithmetic sequence
      provided for each side. The painter charges $1 per digit when
      painting a house number.
34
35    The problem clearly requires us to:
36    1. Identify the numbers on each side of the street.
37    2. Group these numbers by the number of digits (1-digit, 2-digit
      , or 3-digit).
38    3. Calculate the total cost per side of the street, then find
      the sum for both sides.
39
40    The expected solution uses the arithmetic sequence formulas and
      organizes the numbers by digit count. The student's solution
      tries to achieve this goal but deviates significantly in
      implementation.
41
42    ---
43
44    ### **Step 2: Address Lists**
45
46    #### **South Side:**
```

```
47    - Numbers belong to the arithmetic sequence \(4, 10, 16, \dots,
      118\).
48    - Common difference \(d = 6\).
49    - Using the \(n\)-th term formula \(a_n = a + (n-1)d\), the
      \(20^{\text{th}}\) term is:
50    \[
51    a_{20} = 4 + 6(20-1) = 4 + 6(19) = 4 + 114 = 118.
52    \]
53    The numbers are confirmed to be \(4, 10, 16, \dots, 118\).
54
55    #### **North Side:**
56    - Numbers belong to the arithmetic sequence \(3, 9, 15, \dots,
      117\).
57    - Common difference \(d = 6\).
58    - Using the \(n\)-th term formula \(a_n = a + (n-1)d\), the
      \(20^{\text{th}}\) term is:
59    \[
60    a_{20} = 3 + 6(20-1) = 3 + 6(19) = 3 + 114 = 117.
61    \]
62    The numbers are confirmed to be \(3, 9, 15, \dots, 117\).
63
64    The student constructs these lists correctly in their solution.
      **Step 2 is correct.**
65
66    ---
67
68    ### **Step 3: Counting 1-digit, 2-digit, and 3-digit Numbers**
69
70    The student attempts to classify the numbers into groups based
      on digit count. Let's verify the classification for each side
      based on the expected solution:
71
72    #### **South Side:**
73    - **1-digit numbers**: \(4\) (1 number).
74    - **2-digit numbers**: From \(10\) to \(94\), inclusive.
75    Using the \(n\)-th term formula for the first two-digit (\(10\))
       and last two-digit (\(94\)):
76    \[
77    10 = 4 + 6(n-1) \implies n = 2.
78    \]
79    \[
80    94 = 4 + 6(n-1) \implies n = 16.
81    \]
82    So there are \(16 - 2 + 1 = 15\) two-digit numbers.
83    - **3-digit numbers**: From \(100\) to \(118\), inclusive. There
       are \(20 - 16 = 4\) three-digit numbers.
84
85    Thus:
86    - 1-digit: \(1\) number,
87    - 2-digits: \(15\) numbers,
88    - 3-digits: \(4\) numbers.
89
90    #### **North Side:**
91    - **1-digit numbers**: \(3, 9\) (2 numbers).
92    - **2-digit numbers**: From \(15\) to \(99\), inclusive.
93    Using the \(n\)-th term formula for the first two-digit (\(15\))
       and last two-digit (\(99\)):
94    \[
95    15 = 3 + 6(n-1) \implies n = 3.
96    \]
97    \[
98    99 = 3 + 6(n-1) \implies n = 17.
99    \]
```

```
100   So there are \(17 - 3 + 1 = 15\) two-digit numbers.
101   - **3-digit numbers**: From \(105\) to \(117\), inclusive. There
      are \(20 - 17 = 3\) three-digit numbers.
102
103   Thus:
104   - 1-digit: \(2\) numbers,
105   - 2-digits: \(15\) numbers,
106   - 3-digits: \(3\) numbers.
107
108   The student has errors in their count. For example:
109   - They incorrectly list certain numbers as having the wrong
      number of digits.
110   - They double-count or mislabel numbers. (For instance, \(4\) is
       listed as having multiple digit counts.)
111
112   **Step 3 is incorrect** due to significant misclassification.
113
114   ---
115
116   ### **Step 4: Cost Calculation**
117
118   Using the corrected counts:
119
120   #### **South Side:**
121   \[
122   \text{Cost} = (1 \cdot 1) + (2 \cdot 15) + (3 \cdot 4) = 1 + 30
      + 12 = 43.
123   \]
124
125   #### **North Side:**
126   \[
127   \text{Cost} = (1 \cdot 2) + (2 \cdot 15) + (3 \cdot 3) = 2 + 30
      + 9 = 41.
128   \]
129
130   #### **Total Cost:**
131   \[
132   \text{Total} = 43 + 41 = 84.
133   \]
134
135   The student's calculation is incorrect. They arrive at a total
      of \(61\), which significantly underestimates the total due to
      their misclassification of numbers.
136
137   **Step 4 is incorrect** due to incorrect cost calculation.
138
139   ---
140
141   ### **Step 5: Final Verification and Grade**
142
143   - The student correctly identified the arithmetic sequences.
144   - However, they made significant errors in counting the number
      of digits for many house numbers on both the north and south
      sides. These errors carried over into their cost calculation,
      leading to an incorrect total.
145
146   **Verification: Is the answer correct (Yes/No)? No**
```

