# OpenReview forum: "When To Solve, When To Verify: Compute-Optimal Problem Solving and Generative Verification for LLM Reasoning"
_colmweb.org/COLM/2025/Conference — COLM 2025_

### Official Review · Reviewer_CSkR · 2025-05-09

**Rating:** 7
**Confidence:** 4
**Ethics Flag:** 1

**Summary:**

The paper studies the performance of test-time scaling methods under different computation budgets. It conducts extensive experiments and compares two test-time scaling methods:  Self-Consistency (SC) and Generative Reward Models (GenRM). The results show SC performs better with a lower budget, and GenRM can achieve a higher success rate by generating CoT verification tokens. While the result holds across different model sizes and task difficulties, the authors give an empirical conclusion of the optimal ratio of solutions/verification numbers.

**Reasons To Accept:**

- The experiments are solid and sufficient to support the claim. The authors evaluate various models on different tasks and settings to obtain the conclusion.

- The optimal ratio of answer and verification sizes is useful for the community. The authors perform extensive experiments and give the empirical conclusion of balancing the answer and verification size. This result can help further work to obtain better results without further tuning.

**Reasons To Reject:**

The conclusions do not provide much new insight or information. The authors only perform empirical studies to inspect the final performance, without further insight or analysis. More further questions need answer. For example, is there an upper performance bound given enough budget, and how much computation is needed to reach it, or some guidance to improve the performance by limiting each verification length?

---

> ### Author Response · Authors · 2025-06-01
> **Author response to Reviewer CSkR**
>
> We thank the reviewer for their feedback and are pleased they found our study (a) useful to the community & helpful for obtaining better results, and (b) backed by solid & extensive experiments.
>
> **New Insights**\
> We respectfully disagree about limited technicality & highlight several technical nuances & novel insights from our work:
> - Firstly, the original GenRM paper [1] doesn’t release their models & conducts experiments on the relatively simpler GSM8K dataset. In contrast, we reproduce GenRM on the more challenging MATH dataset, which required substantial technical effort. We’ll make our data, models, & code publicly available. Additionally, we’ll release all inference outputs from larger models like QwQ-32B, whose generation is technically demanding. This will be a valuable resource for the community.
> - Overall, our motivation is that as GenRMs gain popularity, it’s important to shed light on their high costs, & encourage research toward making them efficient.
> - As shown in Fig. 1a, prior work [1] shows that GenRM outperforms SC at a fixed number of solutions, giving the false impression that GenRM is more efficient. However, this neglects the high cost of scaling verifications. More prior work [2] used GenRM to double the performance of SC on AIME24, but required 10K LLM calls, costing $650 *per problem*, to do so.
> - Prior work lacked a method for comparing different test-time scaling strategies (such as GenRM & SC) in terms of inference cost. Our work is the first to develop a novel framework for such comparisons.
> - Using this framework, we show that contrary to previous findings, SC outperforms GenRM under most practical budgets. To our knowledge, our work is the first to shed light on the high cost of scaling verifications.
> - Prior work has also not studied the tradeoff between scaling solutions & verifications in GenRM to obtain the best performance for a compute budget. Our work is the first to perform a novel scaling law analysis, & shows that solutions should be scaled more rapidly than verifications.
>
> [1] Generative Verifiers: Reward Modeling as Next-Token Prediction\
> [2] Sample, Scrutinize & Scale: Effective Inference-Time Search by Scaling Verification
>
> **Analysis Beyond Final Performance**\
> We highlight diverse analyses from the paper that go beyond performance:
> - Firstly, we highlight that the focus of our analyses is the computational efficiency of GenRMs, unlike prior works [1,2], which only focus on their performance.
> - In Fig. 3 (left) we show that GenRM benefits difficult problems more. This would help practitioners to choose between SC & GenRM depending on task difficulty.
> - In Fig. 3 (right) we show that a fine-tuned verifier can match the performance of the base verifier with 16x less compute, giving insight into the role of high-quality verifiers, and encouraging practitioners to fine-tune verifiers whenever possible.
> - We also provide a qualitative example in Appendix K, showing how generating a verification CoT improves the final verdict & makes it more interpretable.
>
> [1] Generative Verifiers: Reward Modeling as Next-Token Prediction
> [2] Critique-out-loud Reward Models
>
> **Upper Bound on Performance**
> - We thank the reviewer for the question. The performance upper bound is given by pass@k (i.e., coverage), the probability that at least one of k sampled solutions is correct, assuming an oracle verifier.
> - In https://postimg.cc/2VLHVMHB, Llama 8B achieves 100% pass@k at k=$10^4$, consistent with [1]. Fig. 1b shows that performance saturates for both SC & GenRM: SC peaks at 55% ($2^7$ FLOPs) and GenRM at 59% ($2^{14}$ FLOPs).
>
> |Inference FLOPs|SC|GenRM|Coverage|
> |-|-|-|-|
> |$2^2$|40%|35%|55%|
> |$2^7$|55%|50%|90%|
> |$2^{14}$|55%|59%|100%|
>
> [1] Large Language Monkeys: Scaling Inference Compute with Repeated Sampling
>
> **Impact of verification length**
> - We thank the reviewer for the insightful question. In L159-160, we highlight that the inference compute of GenRM is proportional to the product of the solution & verification lengths. In a hypothetical scenario, assuming fixed solution length & GenRM-FT performance, the reduction in the verification length will make GenRM-FT more compute-efficient.
> - To understand the effect of verification length in this hypothetical scenario, we evaluate the compute required to match SC & compute required to outperform SC by 4% (Figure 1) when the verification length is varied from 2048 to 512:
>
> |Tokens per verification|Compute to match SC|Compute to outperform SC|
> |-|-|-|
> |2048|8x|128x|
> |1024|4x|64x|
> |512|2x|32x|
> |256|1x|16x|
>
> - We find that current verifiers would have to reduce their response length by 8x, while maintaining their performance, to match SC. It remains unclear if it would be possible to achieve such gains in the efficiency of GenRMs.
> - We hope that the shortcomings of current GenRMs highlighted by our work encourage the community to focus on the efficiency of GenRMs. We believe that this is an important direction for future research.

---

> > ### Author Response · Authors · 2025-06-04
> > **Gentle Reminder**
> >
> > Dear Reviewer CSkR,
> >
> > We appreciate your thoughtful comments on our paper. We hope that our rebuttal has addressed your concerns. If you have any further comments or questions, we’d be happy to clarify.
> >
> > Best,
> >
> > Authors of Submission232

---

> > > ### Author Response · Authors · 2025-06-07
> > > **Reminder 2**
> > >
> > > Hi Reviewer CSkR,
> > >
> > > Thank you for handling our paper and for your thoughtful comments. As the author-reviewer discussion period is ending soon, please let us know if you have any further questions or comments—we’d be happy to address them.

---

> > > > ### Comment · Reviewer_CSkR · 2025-06-08
> > > >
> > > > Thanks for the response. The additional results give more insight into the verification vs SC and have addressed most of my concerns. I have raised my rating accordingly and am looking forward to more studies on budget-aware inference.

---

> > > > > ### Author Response · Authors · 2025-06-09
> > > > > **Thanks!**
> > > > >
> > > > > Thank you very much for your encouraging feedback! We will incorporate all responses in the camera-ready version of the paper.

---

> ### Comment · Area_Chair_H7TE · 2025-06-05
>
> Hi Reviewer CSkR! Just a reminder that the discussion period for COLM papers has begun. Could you please take a look at the author response to review and let them know whether it addresses any of your outstanding questions?
>
> Thanks,
> Your AC

---

### Official Review · Reviewer_Kdvi · 2025-05-11

**Rating:** 7
**Confidence:** 3
**Ethics Flag:** 1

**Summary:**

This paper presents an empirical study examining two specific dimensions of test-time scaling in reasoning problems: (i) Sampling multiple solutions, combined with self-consistency voting  (SC) and (ii) Sampling multiple verification chains (with generative reward models/GenRM) for each solution. The authors argue that the computational budget for verification is often ignored in prior work, and that the overall computational budget for both solution generation and verification should be considered. In particular, the authors ask, under a specific computation budget, should one allocate more computation for sampling more solutions or instead sample more verification chains per solution. Through empirical studies on reasoning problems, the authors find that in low computational budget situations, SC leads to better performance, whereas, when more computation is available, scaling GenRM leads to better performance. The authors also present scaling laws for number of solutions and verification chains.

**Questions To Authors:**

* In the context of thinking models that encapsulate the role of verification during the process of solution generation, how are these findings/guidelines for allocating computational budget applicable/relevant?

**Reasons To Accept:**

* The empirical findings are insightful and provide practical guidance on effective test-time scaling.
* Hypotheses are verified with comprehensive evaluation across multiple models, benchmarks and compute regimes.

**Reasons To Reject:**

* Technical component of the paper is somewhat limited. It would have been interesting to see an approach that unlocks the benefits of GenRM at a lower compute regime, such as by fine-tuning small verifiers.
* Although the authors provide a brief discussion on thinking models, they have been excluded from the main analysis, citing that controlling the amount of compute in such models is uncontrollable. However, I think they are an important part of the analysis. For example, with extended thinking, the model encapsulates the role of the RM and it would be interesting to see if it can be better than SC for a given compute budget.
* Although SC seems superior for the tasks considered in this work, its applicability can be limited beyond problems that have a specific, well defined final answer, which makes me question the scope of these conclusions. How are the findings in this paper applicable to problems that have more open-ended responses?

---

> ### Author Response · Authors · 2025-06-01
> **Author response to Reviewer Kdvi**
>
> We thank the reviewer for their feedback and are excited they found our findings (a) insightful, (b) of practical relevance, & (c) supported by extensive experiments.
>
> **Technical Component**\
> We respectfully disagree about limited technicality & highlight several technical nuances & novel insights from our work:
> - Firstly, the original GenRM paper [1] doesn’t release their models & conducts experiments on the relatively simpler GSM8K dataset. In contrast, we reproduce GenRM on the more challenging MATH dataset, which required substantial technical effort. We’ll make our data, models, & code publicly available, and release all outputs from larger models like QwQ-32B, whose inference is technically demanding. This will be a valuable resource for the community.
> - Overall, our motivation is that as GenRMs gain popularity, it’s important to shed light on their high costs, & encourage research toward making them efficient.
> - As shown in Fig. 1a, prior work [1] shows that GenRM outperforms SC at a fixed number of solutions, giving the false impression that GenRM is more efficient. However, this neglects the high cost of scaling verifications. More prior work [2] used GenRM to double the performance of SC on AIME24, but required 10K LLM calls, costing $650 *per problem*, to do so.
> - Prior work lacked a method for comparing different test-time scaling strategies (such as GenRM & SC) in terms of inference cost. **Our work is the first to develop a novel framework for such comparisons.**
> - Using this framework, we show that contrary to previous findings, SC outperforms GenRM under most practical budgets. To our knowledge, our work is the first to shed light on the high cost of scaling verifications.
> - Prior work has also not studied the tradeoff between scaling solutions & verifications in GenRM to obtain the best performance for a compute budget. Our work is the first to perform a **novel scaling law analysis**, & shows that solutions should be scaled more rapidly than verifications.
>
> [1] Generative Verifiers: Reward Modeling as Next-Token Prediction\
> [2] Sample, Scrutinize & Scale: Effective Inference-Time Search by Scaling Verification
>
> **Fine-tuning Smaller GenRMs**
> - We thank the reviewer for this suggestion! Prior work [1-4] used models of the same size for generating solutions & verifications, which is why we studied this setting. The ability to use small verifiers & large generators is an open question, largely unanswered in the literature.
> - We ran an additional experiment on the reviewer's suggestion. Specifically, we fine-tuned LLaMA-8B (small GenRM-FT) on solutions generated by LLaMA-70B. We then compared its performance against LLaMA-70B (large GenRM-base) on solutions from LLaMA-70B on MATH test set here: https://postimg.cc/G4VX9XJ2.
> - We find that SC outperforms the small verifier across the low compute regime, & small GenRM requires 4x compute to match SC. This indicates that while small verifiers are efficient, their performance still lags behind SC for most practical budgets. We’ll add this discussion to the camera-ready.
>
> [1] Generative Verifiers: Reward Modeling as Next-Token Prediction\
> [2] Sample, Scrutinize & Scale: Effective Inference-Time Search by Scaling Verification\
> [3] V-STaR: Training Verifiers for Self-Taught Reasoners\
> [4] Training Verifiers to Solve Math Word Problems
>
> **Analysis of Thinking Models**
> - We believe there might be a misunderstanding. An analysis of thinking models (QwQ-32B) is available in L281-288 of the paper. Fig. 4b shows that our findings hold true for thinking models: on AIME25, GenRM requires 4× more compute to match SC, & achieves a 2.5% improvement with 16× more compute.
> - This highlights that while thinking models can scale test-time compute sequentially in a single CoT, scaling compute parallelly by SC can provide further gains (+12%).
> - Further, these results show that while thinking models are capable of self-reflection, adding a separate verification step by GenRM can further improve performance.
> - Guidance to practitioners: while thinking models can scale test-time compute via longer CoT, scaling solutions & verifications can provide additional gains.
>
> **Relevance in open-ended domains**
> - Most LLM reasoning research focuses on math & science tasks, where final answers are available, like MATH, AIME, & GPQA.
> - Even for coding, benchmarks like LiveCodeBench (LCB) contain test cases with a verifiable answer (i.e., expected output from the code) [1]. This allows us to apply SC, as done by [2].
> - We run an additional experiment on 128 problems from LCB using Llama-3.1-8B here: https://postimg.cc/7GgDB1Kt. We find that GenRM-Base requires ~16x more compute to match SC.
> - We agree that SC might not be applicable in all scenarios. We will add this limitation & the coding results in the camera-ready.
>
> [1] LiveCodeBench: Holistic & Contamination Free Evaluation of Large Language Models for Code\
> [2] Universal Self-Consistency for Large Language Model Generation

---

> > ### Author Response · Authors · 2025-06-04
> > **Gente Reminder**
> >
> > Dear Reviewer Kdvi,
> >
> > We appreciate your thoughtful comments on our paper. We hope that our rebuttal has addressed your concerns. If you have any further comments or questions, we’d be happy to clarify.
> >
> > Best,
> >
> > Authors of Submission232

---

> > > ### Author Response · Authors · 2025-06-07
> > > **Reminder 2**
> > >
> > > Hi Reviewer Kdvi,
> > >
> > > Thank you for handling our paper and for your thoughtful comments. As the author-reviewer discussion period is ending soon, please let us know if you have any further questions or comments—we’d be happy to address them.

---

> > > > ### Author Response · Authors · 2025-06-09
> > > > **Reminder**
> > > >
> > > > Hi Reviewer Kdvi,
> > > >
> > > > Thank you for handling our paper and for your thoughtful comments. As the author-reviewer discussion period is ending soon, please let us know if you have any further questions or comments—we’d be happy to address them.
> > > >
> > > > Best,
> > > > Authors

---

> > > > > ### Comment · Reviewer_Kdvi · 2025-06-11
> > > > > **Follow-up**
> > > > >
> > > > > I thank the authors for the detailed response. The responses addressed my concerns regarding the significance of the findings. I also appreciate the additional experiment with a fine-tuned verifier. I am raising my score.

---

> > > > > > ### Author Response · Authors · 2025-06-11
> > > > > > **Thanks!**
> > > > > >
> > > > > > Thank you very much for your encouraging feedback! We will incorporate all responses in the camera-ready version of the paper.

---

> ### Comment · Area_Chair_H7TE · 2025-06-05
>
> Hi Reviewer Kdvi! Just a reminder that the discussion period for COLM papers has begun. Could you please take a look at the author response to review and let them know whether it addresses any of your outstanding questions?
>
> Thanks,
> Your AC

---

### Official Review · Reviewer_hnKb · 2025-05-21

**Rating:** 6
**Confidence:** 5
**Ethics Flag:** 1

**Summary:**

One is presented with two choices when it comes to scaling test-time compute for reasoning: scale a number of solutions and apply majority vote on top, or scale verification compute via generative verifiers. The paper investigates which choice is best suited under different compute budgets and shows that majority vote or self-consistency is preferred in the low-compute regime, but generative verification can dominate in high-compute regime.

The paper's main contribution are:
1. compute-matched analysis of SC vs GenRM and some interesting findings related to that.
2. scaling law for GenRM: showing that it is best to scale number of sampled solution twice as much as scaling number of verifications.

The authors run experiments on math and scientific reasoning tasks, and the first finding is shown to mostly hold with different model sizes.

**Questions To Authors:**

- How good is you GenRM? could you evaluate it on benchmarks such as ProcessBench (even just using outcome labels) and report results?
- Will your findings translate into a compute-optimal strategy for reasoning with GenRM? If so, how?

**Reasons To Accept:**

- Timely and practical problem: LLM inference cost is real, and test-time optimization matters.
The authors conduct a multitude of experiments to support their finding that majority vote is better at lower inference budgets, and the results are mostly consistent across settings.
- The scaling law for GenRM could be useful but unclear whether it will generalize well.

**Reasons To Reject:**

- It remains unclear what to do with these results. The authors do not propose a compute-optimal strategy to allocate compute to a given input problem. When the authors say "SC is better at low compute, GenRM is better at high compute" I'm inclined to ask what does "low compute" mean here? Obviously low compute means different things in under different model sizes, tasks, etc. In other words, the findings do not translate into a practical recipe to scale inference compute with GenRMs. The paper would substantially benefit from such investigation.
- I'm concerned the results are impacted by the performance of the verifier used. The GenRM verifier used is purely outcome-based, and is trained by the authors. Using LLM-as-a-judge or GenRM base is known to not work well for reasoning. Plus, recent works [1] have shown that process-based verification or GenPRM can work better than GenRM. All this makes me doubt the generalizability of the findings.
- While the findings may be useful in the meantime, it remains unclear whether they will hold in 1-2 years when GenRMs become generally more efficient and better optimized.
- While GenRM indeed seems to catch up later, the improvements from GenRM remain very marginal e.g., +1.7% (MATH), +2.1% (GPQA). One risk here is that readers may come to the understanding that SC is almost always better, which is false since SC will only work in domains/tasks where the correct answer will dominate the incorrect answers. In more difficult or unseen domains, or in domains where there is no final answer to begin with (e.g., code generation or theorem proving) SC is not applicable and the paper findings become of much less value.



[1] https://arxiv.org/abs/2504.16828

---

> ### Author Response · Authors · 2025-05-31
> **Author response to Reviewer hnKb**
>
> We thank the reviewer for their feedback & are motivated that they found our work (a) timely, (b) practical, & (c) extensive.
>
> **GenRM scaling laws**
> - We clarify that our findings about scaling laws are robust across model families (Qwen; Fig. 10) & sizes (Llama 70B; Fig. 11).
>
> **Low/high-compute regimes**
> - Across multiple models & benchmarks (Figs. 1, 4, 5a, 7), SC outperforms GenRM up to ~$2^{10}$ FLOPs. *We term this regime, of less than $2^{10} = 1024$ solutions, as low compute*. Most papers generate solns within this regime [1,2]. We’ll add this to camera-ready.
>
> [1] Beyond Human Data: Scaling Self-Training for Problem-Solving with Language Models.\
> [2]  Smaller, Weaker, Yet Better: Training LLM Reasoners via Compute-Optimal Sampling.
>
> **Performance of our GenRM-FT & ProcessBench**
> - Our GenRM-FT outperforms SC by 4-5.4% (Fig. 1 \& 7). Also, while the original GenRM reports a relative improvement of 2% over SC (Fig. 5c in [1]), our GenRM-FT achieves a relative improvement of 6.9% (Fig. 1b in our draft). This shows that our GenRM-FT is well-trained.
> - GenRM-FT significantly outperforms GenRM-Base (Fig. 3b), & also generalizes to the much harder AIME 24 (Fig. 4).
> - On ProcessBench (w/ outcome labels), our GenRM-FT significantly outperforms GenRM-Base on MATH & OlympiadBench, showing easy-to-hard generalization. We'll add these results to camera-ready.
>
> ||MATH|Olympiad|
> |-|-|-|
> |GenRM-Base|52%|38%|
> |GenRM-FT|87%|80%|
>
> [1] Generative Verifiers: Reward Modeling as Next-Token Prediction.
>
> **Comment on GenRM-Base**
> - GenRM-base is only considered for stronger models like LLama 70B & QwQ-32B, which are top performers on ProcessBench.
> - Finetuning models of this size is difficult with academic budgets, making it common to use them without finetuning, validated by [1], which achieves good performance using GenRM-Base. Hence, studying them in a non-fine-tuned setting is highly relevant.
>
> [1] Sample, Scrutinize, and Scale: Effective Inference-Time Search by Scaling Verification.
>
> **Comments on GenPRMs**
> - We focus on outcome reward models (ORMs), since the original GenRM paper uses ORMs, & they are more widely adopted. We defer more complex strategies like PRM due to the need for extensive process-level data & training complexity.
> - We thank the reviewer for sharing ThinkPRM, but **it is unfair to consider it since ThinkPRM [1] is a non-peer-reviewed paper released after the COLM deadline & lacks a compute-matched analysis against SC.**
> - Further, GenPRMs will incur high computational costs. The inference FLOPs for $N$ beams, $M$ steps per solution, & $L$ CoTs per PRM verification will be $O(NML)$. It’s unclear how well they perform with lower budgets.
>
> [1] Process Reward Models That Think.
>
> **Future Longevity of our work**
> - Our framework & findings will play a crucial role in how practitioners evaluate GenRMs in compute-critical situations.
> - Prior work didn’t consider the cost of verification, suggesting that GenRMs are more efficient than existing methods [1]. As GenRMs gain popularity, shedding light on their high cost is a timely contribution.
> - Moreover, we provide a framework to study the inference cost & scaling behavior of GenRM-based methods, which will still be relevant as GenRMs evolve, & will catalyze progress toward efficient GenRMs.
>
> [1] Generative Verifiers: Reward Modeling as Next-Token Prediction.
>
> **Clarification on SC vs GenRM takeaway**
> - Our motivation is not to claim that GenRMs can never be efficient. Rather, we aim to highlight a shortcoming of current models & spur research toward efficient GenRMs.
> - We don’t suggest that SC is always better than GenRM. GenRM achieves relative improvements of 7% & 8.3% over SC in Figs. 1 & 7, & doubles its performance in Fig. 4a. This is also stated in our takeaway (L248).
>
> **Relevance in open-ended domains**
> - Most LLM reasoning research focuses on math & science tasks, where final answers are available, like MATH, AIME, & GPQA.
> - Even for coding, benchmarks like LiveCodeBench (LCB) contain test cases with a verifiable answer (i.e., expected output from the code) [1]. This allows us to apply SC, as done by [2].
> - We run an additional experiment on 128 problems from LCB using Llama-3.1-8B here: https://postimg.cc/7GgDB1Kt. We find that GenRM-Base requires ~16x more compute to match SC.
> - We agree that SC might not be applicable in all scenarios. We will add this limitation & the coding results in the camera-ready.
>
> [1] LiveCodeBench: Holistic and Contamination Free Evaluation of Large Language Models for Code.\
> [2] Universal Self-Consistency for Large Language Model Generation.
>
> **Compute-optimal strategy for scaling GenRM**
> - Overall, more compute should be spent on scaling solutions vs. verifications.
> - Gains from scaling verification saturate at around 16-32 verifs, while scaling solns up to 256-512 helps.
> - Fig. 9a compares scaling verifs (left) to scaling solns (right), & shows that performance increases more steeply with scaling solns (right)

---

> > ### Author Response · Authors · 2025-06-04
> > **Gentle Reminder**
> >
> > Dear Reviewer hnKb,
> >
> > We appreciate your thoughtful comments on our paper. We hope that our rebuttal has addressed your concerns. If you have any further comments or questions, we’d be happy to clarify.
> >
> > Best,
> >
> > Authors of Submission232

---

> > > ### Author Response · Authors · 2025-06-07
> > > **Reminder 2**
> > >
> > > Hi Reviewer hnKb,
> > >
> > > Thank you for handling our paper and for your thoughtful comments. As the author-reviewer discussion period is ending soon, please let us know if you have any further questions or comments—we’d be happy to address them.

---

> ### Comment · Area_Chair_H7TE · 2025-06-05
>
> Hi Reviewer hnKb! Just a reminder that the discussion period for COLM papers has begun. Could you please take a look at the author response to review and let them know whether it addresses any of your outstanding questions?
>
> Thanks,
> Your AC

---

> ### Comment · Reviewer_hnKb · 2025-06-08
>
> Thank you for the rebuttal. The results on ProcessBench have helped contextualize the findings with GenRM-FT.  After carefully reading your rebuttal, I remain unsure about the long-term impact of these findings. However, I will update my assessment to 6 since I believe the paper could stir nice discussions related to scaling verifier compute and efficient GenRM.

---

> > ### Author Response · Authors · 2025-06-09
> > **Thanks!**
> >
> > Thank you very much for your encouraging feedback! We will incorporate all responses in the camera-ready version of the paper.

---

### Decision · Program_Chairs · 2025-07-08

**Decision:**

Accept

**Comment:**

This paper explores tradeoffs between scaling test-time compute by generating more solutions or by spending extra computation validating existing solutions. In a large-scale empirical evaluation, it reports a trade-off between the two modes: for small compute budgets, it's better to draw many samples; for large budgets, it's better to spend more time verifying. There was an extensive discussion, during which many initial reviewer concerns were addressed. While some reviewers still have questions about the extent to which these findings would generalize to other answer scaling or verification scaling paradigms, there seems to be widespread agreement that the current results are interesting, timely, and likely to be useful to the community.